# GRAPH-AUGMENTED TABULAR TRANSFORMERS: THE SIMPLICITY ADVANTAGE

## ABSTRACT

Transformer models have recently advanced tabular prediction, but they usually treat rows as independent, ignoring that similar instances often share outcomes. Graph augmentation introduces an explicit inductive bias by connecting instances or features and refining embeddings with Graph Neural Networks (GNNs). We present TANGO (Transformers Augmented with Graphs for Tabular Predictions), a large-scale systematic study (to our knowledge, the largest to date) of graph-augmented tabular transformers across 193 datasets (117 classification, 76 regression). Across this benchmark, TANGO not only improves a strong transformer backbone but also surpasses state-of-the-art tabular foundation models (TabPFNv2, TabICL, MITRA) and consistently outperforms classical tree ensembles (CatBoost, XGBoost) in both classification and regression, achieving the most rank-1 wins, lowest average rank, and smallest relative error gaps. Our analysis yields three insights. (1) Graph augmentation consistently improves a strong transformer backbone across diverse tasks. (2) Static graphs outperform dynamic ones, offering better stability and generalization. (3) Within static graphs, frozen embeddings are overall more reliable, consistently outperforming other variants in regression and classification. These results overturn the assumption that dynamic graphs or joint training are always superior, showing instead that schema-anchored graph priors drive generalization: static graphs enforce a stable relational structure, while dynamic ones often introduce instability and risk overfitting.

## 1 INTRODUCTION

Tabular data is one of the most prevalent modalities in machine learning, being behind applications in healthcare, finance, e-commerce, and scientific discovery. Strong baselines for tabular prediction have long come from tree ensembles, with methods such as XGBoost (Chen & Guestrin, 2016) and CatBoost (Prokhorenkova et al., 2018) setting the standard in both research and industry. More recently, transformer-based architectures have narrowed this gap by adapting attention to structured data, while tabular foundation models push further through large-scale pretraining.

FT-Transformer (Gorishniy et al., 2021) introduced feature tokenization with minimal architectural changes, while SAINT (Somepalli et al., 2021) combined row and column-wise attention with contrastive pretraining. In parallel, tabular foundation models have emerged in two broad families: (i) synthetic pretraining with in-context learning (ICL), including TabPFN (Hollmann et al., 2022) and successors such as TabPFNv2 (Hollmann et al., 2025), TabICL (Zhang et al., 2023), and MITRA (Amazon Science, 2025); and (ii) masked-token pretraining, such as TP-BERTa (Yan et al., 2024).

While SAINT attends across rows, and ICL models (e.g., TabPFNv2, TabICL, MITRA) operate over sets of examples, none explicitly construct or maintain a graph of instance similarity. In practice, however, analysts reason by analogy: two loan applicants with nearly identical financial profiles are likely to share default risk, and two students with similar prior grades are likely to perform similarly on an exam. We pursue this complementary direction and show that making such inter-instance similarity explicit through graph priors yields not only consistent improvements, but also performance that surpasses state-of-the-art tabular foundation models.

Graph augmentation offers a natural way to encode such priors. By connecting instances through similarity or shared features and refining representations with Graph Neural Networks (GNNs),

models can capture inter-instance relationships in addition to intra-instance dependencies. Earlier work has explored related ideas, such as TabGNN (Guo et al., 2022), which builds graphs from feature co-occurrence, and BertGCN (Lin et al., 2021), which integrates graphs with pretrained embeddings for text classification. Yet fundamental questions remain: Does graph augmentation consistently improve performance across benchmarks? Should graphs be static or dynamic? Should embeddings be frozen or trained end-to-end? Do these conclusions hold for both regression and classification?

We conduct, to our knowledge, the largest systematic study of graph-augmented tabular transformers to date, spanning 193 datasets (117 classification, 76 regression). TANGO not only improves a strong transformer backbone, but also delivers the *most rank-1 wins*, the *lowest average rank*, and the *smallest relative error gaps* across both regression and classification tasks, surpassing recent foundation models (TabPFNv2, TabICL, MITRA) and consistently outperforming strong tree-based ensembles (CatBoost, XGBoost), which remain the most widely deployed tabular baselines in practice. Most unexpectedly, *static graph topologies provide the strongest inductive bias*. It is the *graph prior itself—anchored in the schema of the dataset—that drives generalization*. Static graphs enforce a consistent relational structure, reducing variance and improving robustness, whereas dynamic similarity graphs tend to introduce instability as embeddings evolve and may overfit to transient patterns. Within static graphs, *frozen* embeddings are the most reliable overall across regression and classification, while end-to-end training remains competitive in classification. These results overturn the assumption that dynamic graphs or joint training are always superior, and highlight a counter-intuitive *simplicity advantage*: stable, schema-anchored graph priors often generalize best.

In summary, we make three contributions: (i) we show that explicit graph priors not only improve a strong transformer backbone but also *consistently outperform classical tree ensembles (CatBoost, XGBoost) and surpass state-of-the-art tabular foundation models (TabPFNv2, TabICL, MITRA)* across classification and regression; (ii) we provide evidence of a simplicity advantage, showing that static graphs consistently outperform dynamic ones and that frozen embeddings are most stable overall while end-to-end training remains competitive in classification; and (iii) we present the largest systematic study of graph augmentation to date, spanning 193 datasets.

## 2 RELATED WORK

### 2.1 TABULAR LEARNING MODELS

Traditional machine learning for tabular data has long been dominated by tree ensembles, including XGBoost (Chen & Guestrin, 2016), CatBoost (Prokhorenkova et al., 2018), and LightGBM (Ke et al., 2017). These models remain strong baselines due to their ability to handle heterogeneous features, missing values, and complex feature interactions.

More recently, researchers have explored deep learning for structured prediction. TabNet (Arik & Pfister, 2021) introduced sequential feature selection using attentive masks. SAINT (Somepalli et al., 2021) combined attention across both rows (instances) and columns (features), along with contrastive pretraining. FT-Transformer (Gorishniy et al., 2021) showed that a relatively simple transformer with feature tokenization can already achieve competitive performance. While influential, these models are trained from scratch on each dataset and do not exploit large-scale pretraining or in-context learning.

### 2.2 TABULAR FOUNDATION MODELS

A recent line of work pretrains *tabular foundation models* once—typically on large collections of *synthetic* datasets—and reuses them across tasks, either via in-context learning (ICL) or light fine-tuning.

• Synthetic pretraining with in-context learning (ICL): TabPFN (Hollmann et al., 2022) pioneered synthetic pretraining for tabular ICL, training a transformer on millions of synthetic datasets to approximate posterior-like predictions in a single forward pass. Its successor, TabPFNv2 (Hollmann et al., 2025), introduces architectural and scaling refinements that strengthen accuracy and robustness on small-to-medium datasets. TabICL (Zhang et al., 2023) advances the ICL family with a two-stage *column-then-row* attention scheme that builds fixed-dimensional row embeddings before

a lightweight ICL transformer, enabling efficiency on larger tables (tens to hundreds of thousands of rows). MITRA (Amazon Science, 2025) complements these approaches by pretraining on *mixtures of synthetic priors* (e.g., causal and tree-based generators) to promote robust cross-dataset transfer.

• Masked-token (BERT-style) pretraining: In parallel, TP-BERTa (Yan et al., 2024) extends masked-token pretraining to tabular data through a tokenization scheme that encodes values by their relative magnitude within each feature distribution for numerics, and intra-feature attention to fuse feature names and values. Unlike ICL models, TP-BERTa is typically fine-tuned per task, providing a complementary route to transfer on tables.

Together, these models trace two complementary trajectories for tabular FMs: (i) *synthetic-pretraining with ICL* (TabPFN/TabPFNv2, TabICL, MITRA) and (ii) *masked-token pretraining* (TP-BERTa). Our study is complementary to foundation-model approaches: we show that *explicit graph priors* consistently improve a strong transformer backbone (FT-Transformer). While we have not evaluated graph augmentation on top of TabPFN, TabPFNv2, TabICL, MITRA, or TP-BERTa, our results suggest that these priors could further benefit such pretraining paradigms.

### 2.3 Graph Neural Networks for Tabular Data

Graph Neural Networks (GNNs) are effective in domains with inherent relational structure, such as molecules (Gilmer et al., 2017), citation networks (Kipf & Welling, 2017b), and social graphs. Their application to tabular learning is relatively recent. TabGNN (Guo et al., 2022) constructs graphs from feature co-occurrence heuristics and learns embeddings via message passing, but does not exploit transformer or attention-based architectures, which have become the state of the art for tabular representation learning. BertGCN (Lin et al., 2021) integrates graphs with pretrained embeddings, but its focus is primarily on text classification rather than general tabular domains. More recently, MPCFIN models cross-feature interactions with a multiplex GNN to better capture heterogeneous feature relations (Ye et al., 2024). Li et al. provide a survey that organizes these developments into a taxonomy of graph formulations, construction methods, and training strategies for tabular data (Li et al., 2024).

### 2.4 Similarity-Based Inductive Biases

Instance similarity has long been exploited as a powerful inductive bias in machine learning. k-nearest neighbors (Cover & Hart, 1967) and Gaussian processes (Rasmussen & Williams, 2006) directly use similarity to regularize predictions, while semi-supervised learning has leveraged graph-based smoothness assumptions (Zhu et al., 2003). Beyond kNN and Gaussian processes, recent methods continue to exploit similarity, e.g., metric or contrastive learning, and RF-GNN extends this line by using Random Forest proximities to form instance graphs (Farokhi et al., 2024).

## 3 Methodology

Our approach augments transformer-based tabular models with graph neural networks (GNNs), enabling explicit modeling of inter-instance similarity. We describe the base transformer backbone, our graph construction strategies, and the training paradigms.

### 3.1 Base Transformer Backbones

We adopt FT-Transformer (Gorishniy et al., 2021) as a representative backbone. While not the strongest performer compared to recent foundation models—such as TabPFN (Hollmann et al., 2022), TabPFNv2 (Hollmann et al., 2025), TabICL (Zhang et al., 2023), MITRA (Amazon Science, 2025), and TP-BERTa (Yan et al., 2024)—it is widely used as a baseline in tabular deep learning, conceptually simple, and provides a reproducible testbed for isolating the effect of graph augmentation. Our method is model-agnostic and could be applied to stronger backbones (e.g., SAINT (Somepalli et al., 2021) or TP-BERTa (Yan et al., 2024)); here we focus on FT-Transformer to cleanly attribute gains to graph augmentation and leave integration with foundation models to future work. Given $n$ instances and $m$ features, the backbone produces embeddings $\mathbf{H} \in \mathbb{R}^{n \times d}$ that serve as input for graph construction. We denote the resulting graph-augmented model as **TANGO**.

---

**Algorithm 1** Dynamic Graph Construction via $k$-NN

---

**Require:** Embeddings $\mathbf{X} \in \mathbb{R}^{n \times d}$, number of neighbors $k$
**Ensure:** Adjacency matrix $A \in \mathbb{R}^{n \times n}$
 1: **for** $i \leftarrow 1, \ldots, n$ **do**
 2:     Compute cosine similarity between instance $i$ and all others
 3:     Identify top-$k$ nearest neighbors
 4:     Set $A[i, j] = 1$ if $j$ is among top-$k$ neighbors, else 0
 5: **end for**
 6: **return** $A$

---

**Algorithm 2** Static Bipartite Graph Construction

---

**Require:** Tabular data $D \in \mathbb{R}^{n \times m}$ with categorical and continuous features
**Ensure:** Adjacency matrix $A$
 Each categorical feature–value pair is represented by a unique node $(f, v)$
 1: **for** categorical feature $f$ **do**
 2:     **for** each instance $i$ **do**
 3:         Connect $i$ to node $(f, v)$ where $v$ is the value of $f$ in instance $i$
 4:     **end for**
 5: **end for**
 6: **for** continuous feature $g$ **do**
 7:     Normalize values; connect instances to $g$ with weighted edges
 8: **end for**
 9: **for** pairs of categorical features **do**
10:     Compute PPMI and update edges between feature nodes
11: **end for**
12: **return** $A$

---

### 3.2 GRAPH CONSTRUCTION

We study two complementary strategies for constructing graphs over instances: dynamic similarity graphs and static bipartite graphs.

**Dynamic graphs.** In line with similarity-based learning traditions (Cover & Hart, 1967; Rasmussen & Williams, 2006), we construct dynamic graphs using a $k$-Nearest Neighbors ($k$-NN) approach (Algorithm 1): Each instance node $i$ is connected to its top $k$ most similar neighbors. This method dynamically captures instance-based relationships, effectively reflecting natural clusters and local neighborhoods within the data. The adjacency matrix is dynamically recalculated as the embeddings evolve during training, thus adapting the graph structure to the learned embedding space.

**Static bipartite graphs.** Inspired by BertGCN (Lin et al., 2021), we also construct static bipartite graphs connecting instance nodes to categorical and continuous feature nodes (see Algorithm 2). For categorical features, each distinct category is represented as a node, with binary edges indicating the presence of the feature. Continuous features are represented as nodes, and each instance is connected to them by a weighted edge whose weight reflects the rescaled value of that feature for the instance. Feature-feature relationships, specifically between categorical nodes, are captured using Positive Pointwise Mutual Information (PPMI), defined as:

$$\text{PPMI}(i, j) = \max \left( \log \frac{p(i, j)}{p(i)p(j)}, 0 \right) \tag{1}$$

where $p(i, j)$ is the joint probability that features $i$ and $j$ occur together, computed by the frequency of co-occurrences of features $i$ and $j$ in all instances, normalized by the total number of instances. The marginal probabilities $p(i)$ and $p(j)$ are calculated similarly by the frequency of occurrences of individual features $i$ and $j$, respectively, normalized by the total number of instances.

## 3.3 Graph Neural Networks and Training Paradigms

Once the graphs are constructed, we train Graph Neural Networks (GNNs) to aggregate and refine node representations. We evaluate several widely used architectures for relational learning, including Graph Convolutional Networks (GCN) (Kipf & Welling, 2017a), Graph Attention Networks (GAT) (Veličković et al., 2018), Graph Isomorphism Networks (GIN) (Xu et al., 2019), Transformer-based GNNs (TransformerConv) (Shi et al., 2021), GATv2 (Brody et al., 2022), EdgeConv (Wang et al., 2019), Principal Neighborhood Aggregation (PNA) (Corso et al., 2020), and Topology Adaptive Graph Convolutional Networks (TAGConv) (Du et al., 2017).

Hyperparameters are tuned using Optuna, with search spaces covering the number of layers, hidden dimensions, learning rate, weight decay, dropout rate, attention heads (for GAT-style models), and neighborhood size in $k$-NN graphs. Full search spaces and fixed training knobs for all models are provided in Appendix A (Classification) and Appendix B (Regression).

We consider two training paradigms:

- **Joint Training**: The embedding model and GNN are trained end-to-end, so that embeddings and, in the case of dynamic graphs, the graph structure itself evolve together during training.
- **Two-Stage Training**: Embeddings are first learned independently by a backbone tabular model and then frozen; the GNN is trained on top of these fixed embeddings, decoupling representation learning from graph aggregation.

These architectures and training paradigms define the design space of our approach. We next describe the experimental setup used to evaluate their effectiveness across diverse tabular datasets.

## 4 Experimental Setup

We conduct large-scale experiments to systematically assess the impact of graph augmentation on tabular transformers, comparing against state-of-the-art baselines across a wide range of datasets.

### 4.1 Datasets

We evaluate our approach on a large-scale benchmark suite composed of 76 regression datasets and 117 classification datasets, drawn from both the TP-BERTa benchmark (Yan et al., 2024) and the recently released TabArena collection (Pfisterer et al., 2025). These datasets span a wide range of domains and vary in number of samples, proportion of categorical versus continuous features, dimensionality of the feature space, and target complexity, providing a diverse and challenging testbed for systematic evaluation.

Each dataset is partitioned into three subsets: a training set (80%) for model training and hyperparameter optimization, a validation set (10%) for model selection, and a test set (10%) to evaluate and report final performance metrics. Splits are created using fixed random seeds to ensure reproducibility and consistency in all experiments.

For regression datasets, we report performance using Root Mean Square Error (RMSE) on the test split. For classification datasets, we evaluate using the macro-F1 score, which equally weights each class and is robust to class imbalance.

A complete list of the datasets we use, together with their structural properties, is provided in the Appendix (Tables 15 and 16). For classification, we report size, feature composition, categorical feature statistics, missingness, and class balance. For regression, we additionally include basic target statistics (mean, standard deviation, min, max).

### 4.2 Baselines

We compare against state-of-the-art tabular prediction baselines including tree ensembles, plain transformers, and recent foundation models:

- **XGBoost**: Gradient-boosted trees (Chen & Guestrin, 2016), tuned with Optuna per dataset.

- **CatBoost**: Boosting optimized for categorical data (Prokhorenkova et al., 2018), similarly tuned.

- **FT-Transformer**: A plain transformer baseline (Gorishniy et al., 2021), trained from scratch with Optuna-tuned hyperparameters, without graph augmentation.

- **TabICL (classification only)**: A foundation model based on in-context learning, evaluated only on classification tasks (no regression variant available) (Zhang et al., 2023).

- **TabPFNv2**: The successor to TabPFN, with architectural refinements and broader synthetic pre-training; evaluated in standard zero-shot mode (Hollmann et al., 2025).

- **MITRA**: A foundation model that extends synthetic pretraining with mixed synthetic priors, achieving strong performance across large-scale benchmarks (Amazon Science, 2025).

TP-BERTa (Yan et al., 2024) extended masked-token pretraining to tabular data, but has since been surpassed by more recent foundation models such as TabPFNv2 (Hollmann et al., 2025), TabICL (Zhang et al., 2023), and MITRA (Amazon Science, 2025). Since we already include FT-Transformer as a per-dataset baseline, our experimental comparisons focus on these stronger foundation models.

**Shared protocol (all models).** All methods—*including TANGO and every baseline*—use the same 80/10/10 train/validation/test split (fixed seed) and a single preprocessing pipeline fit on train and applied unchanged to val/test (label-encoded categoricals with a unified `Missing` sentinel; median-imputed and standardized continuous features; date-like columns expanded). Tuned methods (CatBoost, XGBoost, FT-Transformer, and TANGO variants) receive an identical Optuna budget (100 trials) and the same early-stopping policy for neural models, with mixed precision and gradient clipping. Foundation models (TabPFNv2, MITRA; TabICL for classification) run in their standard frozen/zero-shot configurations, and we remove any default row caps so each model ingests the full training split. For binary tasks, both neural and foundation models automatically calibrate the decision threshold on the validation set. When full-batch graph training exceeds memory, *graph-based runs (i.e., TANGO)* uniformly fall back to PyG's `NeighborLoader` with conservative fan-out and batch-size backoff. TANGO's optional PPMI edges are used on small/medium datasets but omitted on larger ones to avoid memory issues; in those cases we fall back to the pure instance–feature bipartite graph. Further implementation details are in App. A.

## 4.3 EVALUATION METRICS

Following prior work on tabular benchmarks (Gorishniy et al., 2021; Hollmann et al., 2022; Pham et al., 2023), we evaluate models using rank-based and gap-based statistics for fair comparison across heterogeneous datasets:

- **Rank-1 wins.** For each dataset, the method with the best performance (highest Macro-F1 in classification, lowest RMSE in regression) is counted as a rank-1 win.

- **Average rank.** Methods are ranked on each dataset according to their performance. The average rank across all datasets provides a measure of consistency.

- **Relative gap.** To compare across metrics and scales, we compute the relative gap between a method $m$ and the best-performing method on a dataset: $\Delta = |s_m - s^*|$, where $s_m$ is the score of $m$ and $s^*$ the best score, and then report $\Delta/s_m$. This measures the fraction of error that could be eliminated if $m$ matched the best method. Lower values indicate greater robustness.

- **Representative improvements.** In Table 1, we additionally show the relative improvement of TANGO over the best alternative on selected datasets ("Imp.%"), to illustrate scenarios where graph augmentation yields large margins or remains close when not rank-1.

## 5 RESULTS

We compare TANGO against tree ensemble methods (CatBoost (Prokhorenkova et al., 2018), XGBoost (Chen & Guestrin, 2016)), tabular foundation models (TabICL (Zhang et al., 2023), TabPFNv2 (Hollmann et al., 2025), MITRA (Amazon Science, 2025)), and the plain FT-Transformer (Gorishniy et al., 2021). Our evaluation (Section 4.3) reports rank-1 wins, average

Table 1: Representative classification results (Macro-F1). Best per row in **bold**. "Imp.%" = relative improvement of TANGO vs best other method. Dataset names truncated to 12 chars.

| Dataset (12ch) | CatBoost | XGBoost | FT-T | MITRA | TabICL | TabPFNv2 | TANGO | Imp.% |
|---|---|---|---|---|---|---|---|---|
| *A. Large gains where TANGO is rank-1* | | | | | | | | |
| train_0472_a... | 0.418 | 0.451 | 0.513 | 0.439 | 0.439 | 0.439 | **0.677** | 31.81 |
| b_depressed | 0.473 | 0.477 | 0.487 | 0.479 | 0.470 | 0.426 | **0.565** | 16.13 |
| Employee_Sat... | 0.408 | 0.458 | 0.435 | 0.494 | 0.391 | 0.324 | **0.572** | 15.60 |
| train_1592_D... | 0.686 | 0.738 | 0.750 | 0.601 | 0.704 | 0.692 | **0.824** | 9.78 |
| *B. Not rank-1, but TANGO remains competitive* | | | | | | | | |
| hazelnut-spre... | 0.872 | 0.908 | 0.395 | 0.924 | **0.968** | 0.952 | 0.940 | -2.89 |
| GiveMeSomeCr... | 0.630 | 0.628 | 0.683 | 0.689 | **0.694** | 0.694 | 0.686 | -1.13 |
| credit_card_... | 0.683 | 0.674 | 0.443 | 0.699 | **0.713** | 0.703 | 0.701 | -1.56 |

Table 2: Aggregate classification results across 117 datasets. Rank-1 wins (higher is better), average rank (lower is better), and average relative gap (lower is better). Best values per column are in **bold**.

| Method | Rank-1 Wins | Avg. Rank | Avg. Rel. Gap |
|---|---|---|---|
| CatBoost | 24 | 3.90 | 0.092 |
| XGBoost | 27 | 3.65 | 0.083 |
| FT-T | 23 | 4.43 | 0.474 |
| TabICL | 34 | 3.25 | 0.051 |
| TabPFNv2 | 35 | 3.09 | 0.062 |
| MITRA | 18 | 3.72 | 0.069 |
| TANGO | **59** | **2.26** | **0.029** |

rank, and relative gaps to the best method. Together these metrics capture both competitiveness (how often a method wins) and robustness (how close it remains to the best when it does not win).

## 5.1 CLASSIFICATION RESULTS

Table 1 reports results on a sample of representative classification datasets. We include both cases where TANGO delivers substantial gains and cases where it remains competitive but is not rank-1. This illustrates how graph augmentation behaves across different benchmarks.

The upper block highlights datasets where TANGO achieves large gains over the strongest baseline. For example, on *train_0472_analcatdata_marketing*, *b_depressed*, and *Employee Satisfaction Index*, TANGO improves Macro-F1 by 15–32% compared to the best alternative. On *train_1592_Diabetes-Data-Set*, TANGO similarly exceeds all baselines by nearly 10%. These examples show that graph augmentation can yield substantial improvements on challenging benchmarks, surpassing even recent state-of-the-art foundation models.

The lower block presents cases where TANGO is not rank-1 but remains highly competitive. On *hazelnut-spread-contaminant-detection*, *GiveMeSomeCredit*, and *credit_card_clients_default*, models such as TabICL or MITRA lead by a small margin (1–3%), yet TANGO consistently narrows the gap and outperforms the plain FT-Transformer baseline. This demonstrates that even when not the best overall, TANGO tends to stay close to the strongest foundation models while delivering stable improvements over its backbone.

While Table 1 highlights individual benchmarks, Table 2 summarizes performance across all 117 classification datasets. TANGO clearly dominates, with 59 rank-1 wins (50.4%), well ahead of TabPFNv2 (35, 29.9%) and TabICL (34, 29.1%). It also achieves the lowest average rank (2.26) and the smallest average relative gap (0.029), demonstrating that it not only wins most often but also remains consistently close to the best method on datasets where it is not rank-1.

Among foundation models, TabPFNv2 and TabICL form the next strongest tier, with comparable wins and relative gaps around 0.05–0.06. Classical ensembles (XGBoost and CatBoost) remain

Table 3: Representative regression results (RMSE). Best result per dataset is shown in **bold**.

| Dataset | CatBoost | XGBoost | TabPFNv2 | MITRA | FT-T | TANGO |
|---------|----------|---------|----------|-------|------|-------|
| analcatdata_michiganacc | 0.203 | 0.171 | 0.195 | 0.192 | 0.182 | **0.107** |
| campus_recruitment | 0.071 | 0.049 | 0.051 | 0.059 | 0.090 | **0.040** |
| COVID19_ECDC | 0.003 | 0.003 | 0.071 | 0.030 | 0.006 | **0.001** |
| airfoil_self_noise | 0.232 | 0.200 | **0.156** | 0.197 | 0.234 | 0.159 |
| miami_housing | 0.266 | 0.259 | **0.246** | 0.269 | 0.284 | 0.261 |
| Concrete_Data | 0.040 | 0.042 | **0.039** | 0.043 | 0.051 | 0.042 |

Table 4: Aggregate regression results across 76 datasets. Best values per column are in **bold**.

| Method | Rank-1 Wins | Avg. Rank | Avg. Rel. Gap |
|--------|-------------|-----------|---------------|
| CatBoost | 8 | 3.92 | 0.199 |
| XGBoost | 9 | 3.28 | 0.213 |
| FT-T | 4 | 4.71 | 0.291 |
| TabPFNv2 | 24 | 2.72 | 0.178 |
| MITRA | 6 | 3.58 | 0.236 |
| TANGO | **25** | **2.49** | **0.141** |

competitive but trail the newer models in both rank and robustness, while the plain FT-Transformer baseline performs the worst, with the highest average gap (0.474).

Across all 117 datasets, the median improvement of TANGO over the best baseline is 0.0%, the 75th percentile is +0.24%, and the 90th percentile exceeds +4.3%. Thus, although a few datasets yield very large gains, most improvements are modest but consistent. TANGO's strength lies in reliability: it wins more often and remains closer to the best across a broad range of benchmarks. Prior work has also observed that margins between leading methods are often extremely small on tabular benchmarks. For example, the results in Table E.1 of TabArena (Pfisterer et al., 2025) show that the top two models on *Amazon Employee Access* differ by only 0.001 in ROC AUC, and similar gaps are observed on datasets such as *APSFailure* and *Diabetes130US*. Our findings are consistent with this pattern. TANGO distinguishes itself by securing substantially more rank-1 wins and lower error gaps across the board, even if absolute margins on individual datasets are small.

## 5.2 REGRESSION RESULTS

Table 3 shows regression results on representative datasets. TANGO achieves strong improvements on tasks such as *analcatdata_michiganacc* and *campus_recruitment*, and in some cases dramatically outperforms tree-based and foundation models (e.g., *COVID19_ECDC*). On benchmarks like *airfoil_self_noise*, *miami_housing*, and *Concrete_Data*, TANGO remains close to TabPFNv2 while clearly improving over the FT-Transformer baseline, underscoring its robustness. Unlike the classification excerpt (Table 1), we do not report per-dataset improvement percentages ("Imp.%") for regression tasks; because regression results are more balanced across methods, aggregate statistics (rank-1 wins, average rank, and relative gap) provide a fairer summary of performance.

Table 4 aggregates the results in the regression datasets. TANGO and TabPFNv2 are nearly tied in rank-1 wins (25 vs. 24), but TANGO achieves the lowest average rank (2.49) and the smallest relative gap (0.14). This shows that while the regression results are more balanced than the classification, TANGO often wins, and when it does not, it stays closer to the best baseline.

## 5.3 ABLATION STUDY

To better understand the contributions of different graph construction strategies and embedding training regimes, we conduct an ablation study of TANGO. We focus on two key design axes: (i) whether the graph structure is *static* (precomputed once) or *dynamic* (recomputed adaptively), and (ii) whether tabular embeddings are trained *end-to-end* or *frozen*. We name the four variants as follows: **TANGO-SF** (Static + Frozen), **TANGO-SE** (Static + End-to-end), **TANGO-DE** (Dynamic + End-to-end), and **TANGO-DF** (Dynamic + Frozen).

| Variant | Rank-1 Wins | Rank-1 Wins (%) | Avg. Rank |
|---|---|---|---|
| **Classification (117 datasets)** | | | |
| TANGO-SF (Static + Frozen) | 55 | 47.4% | 2.09 |
| TANGO-SE (Static + End-to-end) | 52 | 44.4% | 2.09 |
| TANGO-DE (Dynamic + End-to-end) | 42 | 35.9% | 2.23 |
| TANGO-DF (Dynamic + Frozen) | 32 | 27.4% | 2.44 |
| **Regression (76 datasets)** | | | |
| TANGO-SF (Static + Frozen) | 27 | 35.5% | 2.29 |
| TANGO-DF (Dynamic + Frozen) | 21 | 27.6% | 2.49 |
| TANGO-SE (Static + End-to-end) | 15 | 20.0% | 2.45 |
| TANGO-DE (Dynamic + End-to-end) | 13 | 17.1% | 2.75 |
| **Overall (193 datasets)** | | | |
| TANGO-SF (Static + Frozen) | 82 | 42.7% | 2.17 |
| TANGO-SE (Static + End-to-end) | 67 | 34.9% | 2.23 |
| TANGO-DE (Dynamic + End-to-end) | 55 | 28.5% | 2.44 |
| TANGO-DF (Dynamic + Frozen) | 53 | 27.5% | 2.46 |

Table 5: TANGO ablation (static/dynamic; end-to-end/frozen). TANGO-SF is most robust overall.

For each dataset, we rank the four variants (rank 1 = best, ties allowed), and report the number of rank-1 wins, the percentage of datasets where each method achieved rank-1, and the average rank across datasets. Results are shown in Table 5.

**Findings.** On classification tasks, **TANGO-SF** (Static + Frozen) achieved the most wins (55) with an average rank of 2.09, closely followed by **TANGO-SE** (Static + End-to-end) with 52 wins. On regression tasks, however, **TANGO-SF** clearly dominated with 27 wins and the best average rank (2.29), indicating that freezing embeddings stabilizes training and improves generalization in continuous targets. Dynamic graph variants consistently underperformed. In conclusion, across all 193 datasets, **TANGO-SF** (Static + Frozen) achieved the highest overall win rate (42.7%) and the best average rank (2.17). These results reinforce two key insights: (i) static graphs provide a more stable inductive bias than dynamic graphs, and (ii) freezing embeddings is particularly advantageous for regression tasks.

## 6 LIMITATIONS AND FUTURE WORK

Our study yields two main takeaways. First, explicit graphs that capture inter-instance similarity reliably improve transformer backbones for tabular data, often surpassing state-of-the-art foundation models. Second, a simplicity advantage emerges: static, schema-anchored graphs with frozen embeddings deliver the most stable and reliable gains, overturning the assumption that dynamic graphs or joint training are always superior.

Despite these results, several limitations remain. Static bipartite graphs can be memory-intensive for high-cardinality features or very large datasets, and scaling to millions of rows remains challenging. Our experiments also focused on FT-Transformer; applying TANGO to stronger backbones such as SAINT, TP-BERTa, or other foundation models may reveal further benefits. While dynamic similarity graphs underperformed in this study, future work could explore hybrid approaches. Similarly, moving beyond simple cosine distance toward richer similarity metrics could make dynamic graphs more stable and effective. Looking ahead, opportunities lie in improving scalability, exploring hybrid static–dynamic graphs, and integrating with tabular foundation models.

## REPRODUCIBILITY STATEMENT

We have made extensive efforts to ensure reproducibility. All datasets used in this study are publicly available. Implementation details, and hyperparameter search spaces are provided in the Appendix. Our code will be released upon publication to further facilitate replication of our results.

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

# A    IMPLEMENTATION DETAILS (CLASSIFICATION)

## A.1    PREPROCESSING

**Splits.** Each dataset is partitioned once into an 80/10/10 train/validation/test split with a fixed seed. The same indices are reused by *all* models.

**Single pipeline fit on train only.** A shared preprocessing pipeline is fit on the training split and applied unchanged to validation/test: (i) categorical columns are label-encoded with a unified `Missing` sentinel (unseen tokens on val/test map to `Missing`); (ii) continuous features receive column-wise median imputation followed by standardization; (iii) date-like fields are expanded into {month, day, hour, minute, second} and the raw field is dropped.

**Tuning budget and early stopping.** Tuned methods (CatBoost, XGBoost, FT-Transformer, and all TANGO variants) use Optuna (TPE) with 100 trials per dataset. Neural models use mixed precision, gradient clipping, `ReduceLROnPlateau`, and early stopping with patience 50.

**Binary threshold selection.** For neural models (FT/TANGO), the decision threshold is tuned on the validation split over a small grid $\{0.05, 0.10, \ldots, 0.95\}$ to maximize Macro-F1, then fixed for test. Foundation baselines also automatically calibrate the decision thresholds.

**Foundation models (row-caps removed).** TabPFNv2, TabICL (classification only), and MITRA are run via AutoGluon (Erickson et al., 2020) in frozen/zero-shot mode and without HPO. We explicitly remove internal row caps so the full training split is ingested (implemented as either `max_rows=None` or a very large cap to disable gating).

**OOM-safe graph policy.** If full-batch message passing exceeds GPU memory, we fall back to PyG's `NeighborLoader` with conservative fan-out and batch backoff; the same policy is used for all TANGO variants.

## A.2    BACKBONE, GRAPHS, AND HEADS

**FT-Transformer backbone.** We use FT-Transformer (Gorishniy et al., 2021) as the row encoder; it returns a $d_{\text{out}}$-dimensional instance embedding. The classification head uses class-weighted cross-entropy with label smoothing or focal loss (weights from train frequencies). Tuned hyperparameters appear in Tab. 6.

**Static bipartite graphs (TANGO-S*).** We build a static instance–feature bipartite graph once per dataset: each instance connects to one feature node per categorical value and to one per continuous column. Positive PMI (PPMI) edges between categorical nodes are added for small/medium datasets when the memory fits; otherwise, we fall back to the bipartite structure alone. Embeddings are either *frozen* (two-stage; "SF") or trained *end-to-end* with the graph head ("SE").

**Dynamic similarity graphs (TANGO-D*).** We form a cosine $k$NN graph over instance embeddings (no self-loops). With *frozen* FT embeddings ("DF") we build the graph once; with *end-to-end* ("DE") we refresh it periodically as embeddings evolve. The $k$ range is tuned (Tab. 7).

**Graph heads.** We evaluate GCN, GAT/GATv2, GIN, TransformerConv, EdgeConv, PNA, and TAG-Conv. Head-level search spaces are in Tab. 7.

## A.3    HYPERPARAMETER SEARCH SPACES (CLASSIFICATION)

FT/TANGO backbone and head search spaces are summarized in Tabs. 6–7. Tree baselines (Cat-Boost, XGBoost) follow Tabs. 8–9.

Table 6: FT-Transformer backbone (classification): tuned hyperparameters.

| Hyperparameter | Distribution | Range / Values | Notes |
|---|---|---|---|
| $d_{\text{out}}$ | categorical | $\{64, 128\}$ | Embedding dim |
| $n_{\text{blocks}}$ | integer | $[1, 2]$ | Transformer blocks |
| $d_{\text{block}}$ | categorical | $\{128, 256\}$ | Hidden size |
| attention_n_heads | categorical | $\{4\}$ | Fixed in code |
| attention_dropout | uniform | $[0.10, 0.30]$ | |
| ffn_d_hidden_mult | uniform | $[1.0, 1.6]$ | |
| ffn_dropout | uniform | $[0.10, 0.30]$ | |
| residual_dropout | uniform | $[0.10, 0.30]$ | |
| loss | categorical | $\{\text{CE+LS, Focal}\}$ | LS $\in [0, 0.10]$; Focal $\gamma \in [1.0, 2.5]$ |
| lr | log-uniform | $[10^{-4}, 5\times10^{-4}]$ | Adam; weight_decay $[10^{-6}, 10^{-4}]$ |

Table 7: TANGO graph head (classification): tuned hyperparameters (applies to static/dynamic variants; arch-specific notes at bottom).

| Hyperparameter | Distribution | Range / Values | Notes |
|---|---|---|---|
| topK (dynamic only) | integer | $[3, 20]$ | Capped at $\sqrt{N}$ |
| num_layers | integer | $[2, 4]$ | |
| hidden_channels | categorical | $\{64, 128, 256\}$ | |
| dropout | uniform | $[0.30, 0.70]$ | |
| lr_ft (joint only) | log-uniform | $[10^{-5}, 5\times10^{-4}]$ | |
| lr_gnn | log-uniform | $[10^{-4}, 10^{-2}]$ | |
| weight_decay | log-uniform | $[10^{-6}, 10^{-3}]$ | |
| Arch-specific | fixed / integer | (per layer) | GAT/GATv2 heads $\in \{1, 2, 4\}$; TAGConv $K \in [3, 10]$; PNA aggregators $\{\text{mean, max, min, std}\}$ |

# B  IMPLEMENTATION DETAILS (REGRESSION)

This section complements Sec. 4.3 for regression (RMSE). Where details are identical to the classification appendix (App. A), we omit repetition and note only the differences.

## B.1  WHAT DIFFERS FROM CLASSIFICATION

- **Metric and loss.** Model selection uses validation RMSE; we report test RMSE. Neural losses are MSE or SmoothL1 (Tab. 11); there is no threshold tuning.

- **Target handling by dataset family.** TP-BERTa regression uses the target *as provided* (identity). TabArena regression uses a train-only, per-dataset target transform chosen by simple heuristics on skewness and outliers (Tab. 10); the fitted transform is applied unchanged to val/test.

## B.2  PREPROCESSING AND SETUP

**Splits and preprocessing.** Same 80/10/10 split (fixed seed) and the same train-only preprocessing pipeline as App. A.1.

**Tuning and OOM policy.** Tuned methods (CatBoost, XGBoost, FT, TANGO) use Optuna, 100 trials; neural models use mixed precision, gradient clipping, `ReduceLROnPlateau`, patience 50. Graph training falls back to PyG's `NeighborLoader` when memory is tight (as in App. A.1).

**Hardware.** Single NVIDIA A100 (80 GB) GPU.

Table 8: CatBoost (classification): search space and fixed knobs.

| Hyperparameter | Distribution | Range / Values |
|---|---|---|
| max_depth | integer | $[3, 10]$ |
| learning_rate | log-uniform | $[10^{-5}, 1]$ |
| bagging_temperature | uniform | $[0, 1]$ |
| l2_leaf_reg | log-uniform | $[1, 10]$ |
| leaf_estimation_iterations | integer | $[1, 10]$ |
| iterations | fixed | 2000 |
| task_type / devices | fixed | GPU / 0 |
| random_seed | fixed | 42 |

Table 9: XGBoost (classification): search space and fixed knobs.

| Hyperparameter | Distribution | Range / Values |
|---|---|---|
| max_depth | integer | $[3, 10]$ |
| min_child_weight | log-uniform | $[10^{-8}, 10^5]$ |
| subsample | uniform | $[0.5, 1.0]$ |
| learning_rate | log-uniform | $[10^{-5}, 1.0]$ |
| colsample_bylevel | uniform | $[0.5, 1.0]$ |
| colsample_bytree | uniform | $[0.5, 1.0]$ |
| gamma | log-uniform | $[10^{-8}, 10^2]$ |
| $\lambda$ (L2) | log-uniform | $[10^{-8}, 10^2]$ |
| $\alpha$ (L1) | log-uniform | $[10^{-8}, 10^2]$ |
| n_estimators | fixed | 2000 |
| objective | fixed | `binary:logistic` or `multi:softprob` |
| random_state | fixed | 42 |

## B.3 Target Normalization (TabArena only)

We fit a *train-only* target transform per dataset and apply it unchanged to validation/test. The mode is selected via heuristics on robust outlier mass and skewness (Tab. 10). TP-BERTa uses the identity transform.

## B.4 Backbone and Graph Augmentation (Regression)

**FT-Transformer.** Same encoder as classification, with a scalar head trained using MSE or SmoothL1; tuned hyperparameters in Tab. 11.

**TANGO (static/dynamic; frozen/end-to-end).** Identical construction to App. A.2: (i) static instance–feature bipartite graphs with optional PPMI edges , and (ii) dynamic $k$NN graphs over embeddings (cosine, no self-loops). We evaluate GCN, GAT/GATv2, GIN, TransformerConv, Edge-Conv, PNA, and TAGConv. Head-level search is in Tab. 12.

## B.5 Hyperparameter Search Spaces (Regression)

FT/TANGO search mirrors classification with loss/objective changed for RMSE selection; see Tabs. 11–12. Tree baselines (CatBoost, XGBoost) use Tabs. 13–14.

Table 10: Target transform policy for regression: *train-only* choice based on outlier fraction ($p_{\text{out}}$) and skewness of training target.

| Mode | When selected (train only) |
|------|------|
| `identity` | TP-BERTa |
| `zscore` | If $p_{\text{out}} \leq 0.02$ and $|\text{skew}(y)| \leq 1.0$ |
| `winsor_robust` | Mild outliers: $0.02 < p_{\text{out}} \leq 0.05$ and $|\text{skew}(y)| \leq 1.0$ |
| `quantile_normal` | Heavy tails: $p_{\text{out}} > 0.05$ |

Table 11: FT-Transformer (regression): tuned hyperparameters.

| Hyperparameter | Distribution | Range / Values | Notes |
|------|------|------|------|
| $d_{\text{out}}$ | categorical | $\{64, 128\}$ | |
| $n_{\text{blocks}}$ | integer | $[1, 2]$ | |
| $d_{\text{block}}$ | categorical | $\{128, 256\}$ | |
| attention_n_heads | categorical | $\{4\}$ | |
| dropouts (attn/ffn/resid) | uniform | $[0.10, 0.30]$ | |
| ffn_mult | uniform | $[1.0, 1.6]$ | |
| loss | categorical | $\{\text{MSE}, \text{SmoothL1}\}$ | SmoothL1 $\beta \in [0.5, 2.0]$ |
| lr / weight_decay | log-uniform | $[10^{-4}, 5 \times 10^{-4}] / [10^{-6}, 10^{-4}]$ | Adam |

## C   DATASET SUMMARY

We evaluate on datasets drawn from TP-BERTa and TabArena. For transparency and reproducibility, we list all classification and regression datasets along with key structural properties. Dataset names are truncated for readability.

**Reported properties (classification).** For each dataset we report: source (TP-BERTa or TabArena), train size, total features, continuous/categorical feature counts, proportion of categorical features, average categorical cardinality, overall missingness, number of classes, and majority-class fraction.

**Reported properties (regression).** We report the same structural statistics as above (source, train size, features, categorical statistics, missingness) and additionally include target mean, target standard deviation, and target min/max computed on the training split.

Table 15: Summary of classification datasets used (TP-BERTa and TabArena).

| dataset | source | train size | #feat. total | cont. | cat. | cat (%) | avg card. | miss. (%) | # classes | maj. (%) |
|------|------|------|------|------|------|------|------|------|------|------|
| GiveMeSomeCredit | TabArena | 119936 | 10 | 10 | 0 | 0 | 0 | 2 | 2 | 93.3 |
| customer_satisfactio | TabArena | 103954 | 21 | 18 | 3 | 14.3 | 2.3 | 0 | 2 | 54.7 |
| SDSS17 | TabArena | 62521 | 11 | 11 | 0 | 0 | 0 | 0 | 3 | 61 |
| APSFailure | TabArena | 60876 | 170 | 170 | 0 | 0 | 0 | 8.3 | 2 | 98.2 |
| Diabetes130US | TabArena | 57277 | 47 | 11 | 36 | 76.6 | 6.8 | 8.2 | 2 | 91.2 |
| kddcup09_appetency | TabArena | 40074 | 207 | 174 | 33 | 15.9 | 439.5 | 68 | 2 | 98.2 |
| bank-marketing | TabArena | 36203 | 13 | 5 | 8 | 61.5 | 4 | 0 | 2 | 88.4 |
| Amazon_employee_acce | TabArena | 26235 | 9 | 9 | 0 | 0 | 0 | 0 | 2 | 94.1 |
| credit_card_clients_ | TabArena | 24044 | 23 | 23 | 0 | 0 | 0 | 0 | 2 | 77.7 |
| HR_Analytics_Job_Cha | TabArena | 15335 | 12 | 2 | 10 | 83.3 | 19.2 | 8.3 | 2 | 75 |
| in_vehicle_coupon_re | TabArena | 10170 | 24 | 7 | 17 | 70.8 | 6.4 | 4.2 | 2 | 57.2 |
| online_shoppers_inte | TabArena | 9881 | 17 | 15 | 2 | 11.8 | 6.5 | 0 | 2 | 84.5 |
| E-CommereShippingDat | TabArena | 8834 | 10 | 6 | 4 | 40 | 3.2 | 0 | 2 | 59.3 |
| jm1 | TabArena | 8743 | 21 | 21 | 0 | 0 | 0 | 0 | 2 | 81.1 |
| heloc | TabArena | 8400 | 23 | 23 | 0 | 0 | 0 | 0 | 2 | 52 |
| Bank_Customer_Churn | TabArena | 8038 | 10 | 8 | 2 | 20 | 2.5 | 0 | 2 | 79.9 |
| coil2000_insurance_p | TabArena | 7900 | 85 | 80 | 5 | 5.9 | 13.8 | 0 | 2 | 94 |
| train_0356_delta_ele | TP-BERTa | 7643 | 6 | 6 | 0 | 0 | 0 | 0 | 2 | 50.1 |
| train_0284_bank8FM | TP-BERTa | 6564 | 8 | 8 | 0 | 0 | 0 | 0 | 2 | 59.7 |
| train_0292_cpu_small | TP-BERTa | 6564 | 12 | 12 | 0 | 0 | 0 | 0 | 2 | 69.7 |
| train_0312_cpu_act | TP-BERTa | 6564 | 21 | 21 | 0 | 0 | 0 | 0 | 2 | 69.7 |

Table 15: Summary of classification datasets used (TP-BERTa and TabArena).

| dataset | source | train size | #feat. total | cont. | cat. | cat (%) | avg card. | miss. (%) | # classes | maj. (%) |
|---|---|---|---|---|---|---|---|---|---|---|
| train_1512_eye_movem | TP-BERTa | 6089 | 22 | 22 | 0 | 0 | 0 | 0 | 2 | 50 |
| NATICUSdroid | TabArena | 5994 | 98 | 98 | 0 | 0 | 0 | 0 | 2 | 65.3 |
| train_0345_delta_ail | TP-BERTa | 5699 | 5 | 5 | 0 | 0 | 0 | 0 | 2 | 53.3 |
| taiwanese_bankruptcy | TabArena | 5455 | 94 | 94 | 0 | 0 | 0 | 0 | 2 | 96.7 |
| train_1736_combined- | TP-BERTa | 5200 | 12 | 12 | 0 | 0 | 0 | 0 | 2 | 75.2 |
| train_1759_Red–Whit | TP-BERTa | 5200 | 12 | 12 | 0 | 0 | 0 | 0 | 2 | 75.2 |
| train_1413_shill-bid | TP-BERTa | 5060 | 11 | 10 | 1 | 9.1 | 997 | 0 | 2 | 89.7 |
| polish_companies_ban | TabArena | 4731 | 64 | 64 | 0 | 0 | 0 | 1.2 | 2 | 93.3 |
| train_1600_VulNoneVu | TP-BERTa | 4560 | 16 | 16 | 0 | 0 | 0 | 0 | 2 | 99 |
| train_0885_compas-tw | TP-BERTa | 4228 | 13 | 13 | 0 | 0 | 0 | 0 | 2 | 52.7 |
| train_1458_kdd_ipums | TP-BERTa | 4155 | 20 | 20 | 0 | 0 | 0 | 0 | 2 | 50.5 |
| train_1578_kdd_ipums | TP-BERTa | 4155 | 20 | 20 | 0 | 0 | 0 | 0 | 2 | 50.2 |
| train_1692_Gender-Cl | TP-BERTa | 4002 | 7 | 7 | 0 | 0 | 0 | 0 | 2 | 50 |
| Bank_Personal_Loan_M | TP-BERTa | 4001 | 12 | 12 | 0 | 0 | 0 | 0 | 2 | 71 |
| UniversalBank | TP-BERTa | 4001 | 12 | 12 | 0 | 0 | 0 | 0 | 2 | 71 |
| churn | TabArena | 4001 | 19 | 19 | 0 | 0 | 0 | 0 | 2 | 86 |
| train_1898_Personal- | TP-BERTa | 4001 | 12 | 12 | 0 | 0 | 0 | 0 | 2 | 70.4 |
| train_2703_compas-tw | TP-BERTa | 3975 | 11 | 11 | 0 | 0 | 0 | 0 | 2 | 50.1 |
| train_1408_national- | TP-BERTa | 3929 | 16 | 9 | 7 | 43.8 | 10.1 | 0 | 2 | 62.3 |
| students_dropout_and | TabArena | 3546 | 36 | 19 | 17 | 47.2 | 13.5 | 0 | 3 | 50 |
| train_0400_analcatda | TP-BERTa | 3237 | 7 | 7 | 0 | 0 | 0 | 0 | 2 | 76.2 |
| Breast_Cancer | TP-BERTa | 3216 | 15 | 6 | 9 | 60 | 3.3 | 0 | 2 | 84.5 |
| train_0509_pollen | TP-BERTa | 3076 | 5 | 5 | 0 | 0 | 0 | 0 | 2 | 50.3 |
| train_1142_Sick_nume | TP-BERTa | 3019 | 29 | 29 | 0 | 0 | 0 | 0 | 2 | 94.1 |
| NFL | TP-BERTa | 2784 | 17 | 11 | 6 | 35.3 | 807.3 | 0 | 2 | 64.4 |
| splice | TabArena | 2552 | 60 | 0 | 60 | 100 | 4.8 | 0 | 3 | 51.9 |
| train_1201_Gender-Re | TP-BERTa | 2533 | 20 | 20 | 0 | 0 | 0 | 0 | 2 | 50.6 |
| seismic-bumps | TabArena | 2059 | 15 | 11 | 4 | 26.7 | 2.5 | 0 | 2 | 93.8 |
| hazelnut-spread-cont | TabArena | 1917 | 30 | 30 | 0 | 0 | 0 | 0 | 2 | 50.1 |
| Marketing_Campaign | TabArena | 1783 | 25 | 22 | 3 | 12 | 217.3 | 0 | 2 | 85.4 |
| train_0948_Ishwar | TP-BERTa | 1758 | 21 | 17 | 4 | 19 | 3.5 | 0 | 2 | 66.7 |
| train_1006_Titanic | TP-BERTa | 1754 | 3 | 3 | 0 | 0 | 0 | 0 | 2 | 67 |
| train_0437_quake | TP-BERTa | 1736 | 3 | 3 | 0 | 0 | 0 | 0 | 2 | 55.6 |
| train_2304_electrici | TP-BERTa | 1597 | 12 | 12 | 0 | 0 | 0 | 0 | 2 | 50.1 |
| train_2305_electrici | TP-BERTa | 1597 | 12 | 12 | 0 | 0 | 0 | 0 | 2 | 50.1 |
| train_2306_electrici | TP-BERTa | 1597 | 12 | 12 | 0 | 0 | 0 | 0 | 2 | 50.1 |
| train_2308_electrici | TP-BERTa | 1597 | 12 | 12 | 0 | 0 | 0 | 0 | 2 | 50.1 |
| train_2389_airlines_ | TP-BERTa | 1597 | 7 | 4 | 3 | 42.9 | 129 | 0 | 2 | 56.8 |
| train_2390_airlines_ | TP-BERTa | 1597 | 7 | 4 | 3 | 42.9 | 127.3 | 0 | 2 | 56.8 |
| train_2391_airlines_ | TP-BERTa | 1597 | 7 | 4 | 3 | 42.9 | 129.7 | 0 | 2 | 56.8 |
| train_2392_airlines_ | TP-BERTa | 1597 | 7 | 4 | 3 | 42.9 | 127 | 0 | 2 | 56.8 |
| train_2393_airlines_ | TP-BERTa | 1597 | 7 | 4 | 3 | 42.9 | 126.7 | 0 | 2 | 56.8 |
| train_2619_sf-police | TP-BERTa | 1597 | 8 | 6 | 2 | 25 | 632.5 | 0 | 2 | 87.6 |
| train_2620_sf-police | TP-BERTa | 1597 | 8 | 6 | 2 | 25 | 627 | 0 | 2 | 87.6 |
| train_2621_sf-police | TP-BERTa | 1597 | 8 | 6 | 2 | 25 | 633.5 | 0 | 2 | 87.6 |
| train_2622_sf-police | TP-BERTa | 1597 | 8 | 6 | 2 | 25 | 635.5 | 0 | 2 | 87.6 |
| TravelInsurancePredi | TP-BERTa | 1585 | 8 | 4 | 4 | 50 | 2 | 0 | 2 | 64 |
| Is-this-a-good-custo | TabArena | 1368 | 13 | 9 | 4 | 30.8 | 8 | 0 | 2 | 88.5 |
| train_1635_Is-this-a | TP-BERTa | 1368 | 13 | 9 | 4 | 30.8 | 8.2 | 0 | 2 | 88.8 |
| MIC | TabArena | 1347 | 111 | 111 | 0 | 0 | 0 | 8.5 | 8 | 84 |
| bt_dataset_t3 | TP-BERTa | 1307 | 17 | 17 | 0 | 0 | 0 | 0 | 2 | 88.4 |
| Fitness_Club | TabArena | 1189 | 6 | 3 | 3 | 50 | 5 | 0.2 | 2 | 70.1 |
| b_depressed | TP-BERTa | 1132 | 21 | 21 | 0 | 0 | 0 | 0 | 2 | 83.5 |
| piracydataset | TP-BERTa | 1129 | 4 | 1 | 3 | 75 | 705.7 | 0 | 2 | 81.1 |
| BankNoteAuthenticati | TP-BERTa | 1088 | 4 | 4 | 0 | 0 | 0 | 0 | 2 | 55.5 |
| website_phishing | TabArena | 1074 | 9 | 0 | 9 | 100 | 2.8 | 0 | 3 | 52.2 |
| train_0526_colleges_ | TP-BERTa | 917 | 15 | 13 | 2 | 13.3 | 27.5 | 0 | 2 | 69.6 |
| train_0555_socmob | TP-BERTa | 915 | 5 | 1 | 4 | 80 | 9.5 | 0 | 2 | 76.4 |
| qsar-biodeg | TabArena | 838 | 41 | 41 | 0 | 0 | 0 | 0 | 2 | 65.6 |
| maternal_health_risk | TabArena | 810 | 6 | 6 | 0 | 0 | 0 | 0 | 3 | 40.9 |
| credit-g | TabArena | 801 | 20 | 7 | 13 | 65 | 4.2 | 0 | 2 | 70.4 |
| anneal | TabArena | 715 | 38 | 7 | 31 | 81.6 | 2.4 | 0 | 5 | 75.5 |
| train_1564_Mammograp | TP-BERTa | 656 | 5 | 5 | 0 | 0 | 0 | 0 | 2 | 50.9 |
| audit_data | TP-BERTa | 615 | 26 | 26 | 0 | 0 | 0 | 0 | 2 | 60.5 |
| audit_risk | TP-BERTa | 615 | 26 | 26 | 0 | 0 | 0 | 0 | 2 | 60.5 |
| trial | TP-BERTa | 615 | 17 | 17 | 0 | 0 | 0 | 0 | 2 | 63.3 |
| diabetes | TabArena | 609 | 8 | 8 | 0 | 0 | 0 | 0 | 2 | 66.5 |
| train_1592_Diabetes- | TP-BERTa | 609 | 8 | 8 | 0 | 0 | 0 | 0 | 2 | 64.2 |
| blood-transfusion-se | TabArena | 593 | 4 | 4 | 0 | 0 | 0 | 0 | 2 | 76.6 |
| train_1752_Wisconsin | TP-BERTa | 555 | 10 | 10 | 0 | 0 | 0 | 0 | 2 | 66.5 |
| train_0435_strikes | TP-BERTa | 490 | 6 | 6 | 0 | 0 | 0 | 0 | 2 | 50.2 |
| loan_train | TP-BERTa | 482 | 11 | 6 | 5 | 45.5 | 2.2 | 0.7 | 2 | 70.3 |
| train_1742_Loan-Pred | TP-BERTa | 482 | 11 | 6 | 5 | 45.5 | 2.2 | 0.6 | 2 | 67.6 |
| train_0445_arsenic-m | TP-BERTa | 433 | 4 | 4 | 0 | 0 | 0 | 0 | 2 | 95.8 |

Table 15: Summary of classification datasets used (TP-BERTa and TabArena).

| dataset | source | train size | #feat. total | cont. | cat. | cat (%) | avg card. | miss. (%) | # classes | maj. (%) |
|---|---|---|---|---|---|---|---|---|---|---|
| train_0446_arsenic-f | TP-BERTa | 433 | 4 | 4 | 0 | 0 | 0 | 0 | 2 | 85 |
| train_0447_arsenic-f | TP-BERTa | 433 | 4 | 4 | 0 | 0 | 0 | 0 | 2 | 96.5 |
| diabetes_data_upload | TP-BERTa | 406 | 16 | 1 | 15 | 93.8 | 2 | 0 | 2 | 62.3 |
| train_1451_early-sta | TP-BERTa | 406 | 16 | 1 | 15 | 93.8 | 2 | 0 | 2 | 63.3 |
| train_1774_Early-Sta | TP-BERTa | 406 | 16 | 1 | 15 | 93.8 | 2 | 0 | 2 | 63.3 |
| Employee Satisfactio | TP-BERTa | 394 | 12 | 7 | 5 | 41.7 | 81.2 | 0 | 2 | 52.5 |
| train_0419_pm10 | TP-BERTa | 394 | 7 | 7 | 0 | 0 | 0 | 0 | 2 | 50 |
| train_1619_NBA-2k20- | TP-BERTa | 350 | 14 | 4 | 10 | 71.4 | 101.1 | 0 | 2 | 98 |
| Customer_Behaviour | TP-BERTa | 319 | 3 | 2 | 1 | 33.3 | 2 | 0 | 2 | 64.9 |
| new_model | TP-BERTa | 319 | 13 | 13 | 0 | 0 | 0 | 0 | 2 | 61.8 |
| train_0472_analcatda | TP-BERTa | 294 | 32 | 32 | 0 | 0 | 0 | 0 | 2 | 68.4 |
| train_0541_plasma_re | TP-BERTa | 249 | 13 | 10 | 3 | 23.1 | 2.7 | 0 | 2 | 55.4 |
| train_1011_cleve | TP-BERTa | 240 | 13 | 13 | 0 | 0 | 0 | 0 | 2 | 53.8 |
| train_1461_heart-fai | TP-BERTa | 236 | 12 | 12 | 0 | 0 | 0 | 0 | 2 | 66.1 |
| train_0408_pharynx | TP-BERTa | 155 | 10 | 10 | 0 | 0 | 0 | 0 | 2 | 58.7 |
| train_0424_autoPrice | TP-BERTa | 126 | 15 | 15 | 0 | 0 | 0 | 0 | 2 | 65.9 |
| train_0446_newton_he | TP-BERTa | 110 | 3 | 2 | 1 | 33.3 | 11 | 0 | 2 | 50.9 |
| train_0546_analcatda | TP-BERTa | 104 | 3 | 2 | 1 | 33.3 | 3 | 0 | 2 | 63.5 |
| train_1333_ricci_vs_ | TP-BERTa | 95 | 5 | 3 | 2 | 40 | 2.5 | 0 | 2 | 52.6 |
| train_0406_visualizi | TP-BERTa | 92 | 3 | 3 | 0 | 0 | 0 | 0 | 2 | 51.1 |
| train_0124_analcatda | TP-BERTa | 82 | 9 | 4 | 5 | 55.6 | 11.4 | 0 | 2 | 62.2 |

Table 16: Summary of regression datasets used (TP-BERTa and TabArena).

| dataset | source | train size | #feat. total | cont. | cat. | cat ratio | avg card. | miss. frac | target mean | target std | target min | target max |
|---|---|---|---|---|---|---|---|---|---|---|---|---|
| diamonds | TabArena | 43228 | 9 | 6 | 3 | 0.3 | 6.7 | 0 | 0 | 1 | -5.199 | 5.199 |
| physiochemic | TabArena | 36618 | 9 | 9 | 0 | 0 | 0 | 0 | -0 | 1 | -1.266 | 2.17 |
| Food_Deliver | TabArena | 36394 | 9 | 6 | 3 | 0.3 | 442.7 | 0 | 0 | 1 | -1.734 | 2.956 |
| superconduct | TabArena | 17002 | 81 | 81 | 0 | 0 | 0 | 0 | 0.561 | 1.34 | -0.784 | 3.944 |
| houses | TabArena | 16513 | 8 | 8 | 0 | 0 | 0 | 0 | 0 | 1 | -4.327 | 1.831 |
| miami_housin | TabArena | 11038 | 15 | 15 | 0 | 0 | 0 | 0 | 0 | 1.005 | -5.199 | 5.199 |
| train_1591_S | TP-BERTa | 7881 | 24 | 11 | 13 | 0.5 | 786.8 | 0 | 0.01 | 0.027 | 0 | 1 |
| train_1890_E | TP-BERTa | 7523 | 13 | 10 | 3 | 0.2 | 198.3 | 0 | 0.047 | 0.146 | 0 | 1 |
| thyroidDF | TP-BERTa | 7367 | 30 | 7 | 23 | 0.8 | 3.4 | 0 | 0.416 | 0.258 | 0 | 1 |
| train_1872_F | TP-BERTa | 6498 | 3 | 1 | 2 | 0.7 | 30 | 0 | 0.042 | 0.102 | 0 | 1 |
| wine_quality | TabArena | 5200 | 12 | 11 | 1 | 0.1 | 2 | 0 | 0 | 1 | -3.227 | 3.657 |
| wines_SPA | TP-BERTa | 5069 | 10 | 5 | 5 | 0.5 | 221.6 | 0.004 | 2.945 | 0.252 | 1 | 3 |
| real_estate_ | TP-BERTa | 3955 | 9 | 7 | 2 | 0.2 | 20 | 0 | 0.063 | 0.039 | 0 | 1 |
| train_1848_M | TP-BERTa | 3836 | 17 | 5 | 12 | 0.7 | 66 | 0 | 0.203 | 0.14 | 0 | 1 |
| train_1623_G | TP-BERTa | 3823 | 5 | 5 | 0 | 0 | 0 | 0 | 0.041 | 0.032 | 0 | 1 |
| train_1901_N | TP-BERTa | 3674 | 5 | 5 | 0 | 0 | 0 | 0 | 0.054 | 0.061 | 0 | 1 |
| train_1228_P | TP-BERTa | 2364 | 20 | 20 | 0 | 0 | 0 | 0 | 0.646 | 0.156 | 0 | 1 |
| train_1878_C | TP-BERTa | 2056 | 9 | 7 | 2 | 0.2 | 138.5 | 0.313 | 0.017 | 0.071 | 0 | 0.912 |
| train_1222_p | TP-BERTa | 2045 | 19 | 19 | 0 | 0 | 0 | 0 | 0.163 | 0.844 | -1 | 1 |
| train_1869_B | TP-BERTa | 1914 | 5 | 5 | 0 | 0 | 0 | 0 | 0.033 | 0.053 | 0 | 1 |
| train_1331_d | TP-BERTa | 1736 | 3 | 3 | 0 | 0 | 0 | 0 | 5.977 | 0.191 | 5.8 | 6.9 |
| train_1769_F | TP-BERTa | 1658 | 5 | 5 | 0 | 0 | 0 | 0 | 0.047 | 0.053 | 0 | 1 |
| train_1594_S | TP-BERTa | 1592 | 13 | 10 | 3 | 0.2 | 789.7 | 0 | 0.547 | 0.16 | 0 | 1 |
| train_1624_A | TP-BERTa | 1230 | 6 | 5 | 1 | 0.2 | 84 | 0 | 0.23 | 0.173 | 0 | 1 |
| Another-Data | TabArena | 1221 | 7 | 6 | 1 | 0.1 | 3 | 0 | -0 | 1 | -3.165 | 1.303 |
| train_1900_A | TP-BERTa | 1221 | 7 | 6 | 1 | 0.1 | 3 | 0 | 0.705 | 0.227 | 0.047 | 1 |
| airfoil_self | TabArena | 1192 | 5 | 5 | 0 | 0 | 0 | 0 | 0 | 1 | -3.098 | 2.337 |
| train_1417_i | TP-BERTa | 1163 | 32 | 24 | 8 | 0.2 | 3.6 | 0 | 0.245 | 0.212 | 0 | 1 |
| healthcare_i | TabArena | 1062 | 6 | 3 | 3 | 0.5 | 2.7 | 0 | -0 | 1.019 | -5.199 | 5.199 |
| train_1874_G | TP-BERTa | 976 | 8 | 5 | 3 | 0.4 | 572 | 0 | 0.007 | 0.028 | 0 | 0.46 |
| train_1449_g | TP-BERTa | 945 | 18 | 15 | 3 | 0.2 | 4.7 | 0 | 0.568 | 0.195 | 0 | 0.986 |
| train_1528_1 | TP-BERTa | 945 | 18 | 15 | 3 | 0.2 | 4.7 | 0 | 0.568 | 0.195 | 0 | 0.986 |
| train_0149_s | TP-BERTa | 915 | 5 | 1 | 4 | 0.8 | 9.5 | 0 | 0.044 | 0.1 | 0 | 1 |
| concrete_com | TabArena | 818 | 8 | 8 | 0 | 0 | 0 | 0 | 0 | 1 | -2.038 | 2.838 |
| train_0925_C | TP-BERTa | 818 | 8 | 8 | 0 | 0 | 0 | 0 | 0.417 | 0.21 | 0.012 | 1 |
| train_1640_C | TP-BERTa | 818 | 8 | 8 | 0 | 0 | 0 | 0 | 0.417 | 0.21 | 0.012 | 1 |
| train_2644_c | TP-BERTa | 818 | 8 | 8 | 0 | 0 | 0 | 0 | 0.417 | 0.21 | 0.012 | 1 |
| train_1564_C | TP-BERTa | 804 | 8 | 8 | 0 | 0 | 0 | 0 | 0.412 | 0.203 | 0 | 1 |
| train_2168_I | TP-BERTa | 801 | 18 | 14 | 4 | 0.2 | 2.5 | 0 | 0.338 | 0.21 | 0 | 1 |
| mechanical_a | TP-BERTa | 739 | 10 | 9 | 1 | 0.1 | 10 | 0 | 4.078 | 2.22 | 2 | 9 |
| QSAR_fish_to | TabArena | 722 | 6 | 6 | 0 | 0 | 0 | 0 | -0 | 1 | -2.705 | 3.732 |
| train_1464_d | TP-BERTa | 594 | 19 | 12 | 7 | 0.4 | 494.1 | 0 | 0.418 | 0.205 | 0 | 1 |
| train_1712_R | TP-BERTa | 545 | 20 | 18 | 2 | 0.1 | 21 | 0.018 | 0.258 | 0.223 | 0 | 1 |
| ph-data | TP-BERTa | 516 | 3 | 3 | 0 | 0 | 0 | 0 | 0.499 | 0.31 | 0 | 1 |
| train_1260_o | TP-BERTa | 504 | 9 | 7 | 2 | 0.2 | 3 | 0 | 0.011 | 0.087 | 0 | 1 |

Table 16: Summary of regression datasets used (TP-BERTa and TabArena).

| dataset | source | train size | #feat. total | cont. | cat. | cat ratio | avg card. | miss. frac | target mean | target std | target min | target max |
|---|---|---|---|---|---|---|---|---|---|---|---|---|
| train_0259_s | TP-BERTa | 490 | 6 | 6 | 0 | 0 | 0 | 0 | 0.042 | 0.071 | 0 | 1 |
| train_0237_a | TP-BERTa | 433 | 4 | 4 | 0 | 0 | 0 | 0 | 0.011 | 0.084 | 0 | 1 |
| train_0251_a | TP-BERTa | 433 | 4 | 4 | 0 | 0 | 0 | 0 | 0.01 | 0.083 | 0 | 1 |
| train_1065_h | TP-BERTa | 408 | 19 | 19 | 0 | 0 | 0 | 0 | 0.092 | 0.104 | 0 | 1 |
| train_0911_f | TP-BERTa | 405 | 12 | 10 | 2 | 0.2 | 9.5 | 0 | 0.009 | 0.024 | 0 | 0.195 |
| train_0258_n | TP-BERTa | 394 | 7 | 7 | 0 | 0 | 0 | 0 | 0.475 | 0.143 | 0 | 0.881 |
| train_0226_a | TP-BERTa | 372 | 3 | 3 | 0 | 0 | 0 | 0 | 0.349 | 0.208 | 0 | 1 |
| User Knowled | TP-BERTa | 321 | 4 | 4 | 0 | 0 | 0 | 0 | 0.452 | 0.267 | 0 | 1 |
| train_0907_U | TP-BERTa | 316 | 32 | 15 | 17 | 0.5 | 2.5 | 0 | 0.525 | 0.231 | 0 | 1 |
| train_1267_a | TP-BERTa | 313 | 5 | 5 | 0 | 0 | 0 | 0 | 0.387 | 0.207 | 0 | 1 |
| train_0211_a | TP-BERTa | 294 | 3 | 1 | 2 | 0.7 | 9 | 0 | 0.56 | 0.219 | 0 | 1 |
| train_1118_j | TP-BERTa | 289 | 17 | 17 | 0 | 0 | 0 | 0 | 0.124 | 0.137 | 0 | 1 |
| train_1845_P | TP-BERTa | 281 | 6 | 6 | 0 | 0 | 0 | 0 | 0.513 | 0.14 | 0 | 1 |
| train_0272_k | TP-BERTa | 250 | 11 | 8 | 3 | 0.3 | 3.3 | 0 | 0.199 | 0.238 | 0 | 1 |
| train_0273_k | TP-BERTa | 250 | 11 | 8 | 3 | 0.3 | 3.3 | 0 | 0.097 | 0.141 | 0 | 0.763 |
| train_0274_k | TP-BERTa | 250 | 11 | 8 | 3 | 0.3 | 3.3 | 0 | 0.093 | 0.141 | 0 | 0.605 |
| train_0279_k | TP-BERTa | 250 | 11 | 8 | 3 | 0.3 | 3.3 | 0 | 0.089 | 0.142 | 0 | 1 |
| train_0235_p | TP-BERTa | 249 | 13 | 10 | 3 | 0.2 | 2.7 | 0 | 0.273 | 0.137 | 0 | 1 |
| train_0130_b | TP-BERTa | 224 | 9 | 3 | 6 | 0.7 | 2.7 | 0 | 0.488 | 0.203 | 0 | 1 |
| train_1787_L | TP-BERTa | 196 | 15 | 8 | 7 | 0.5 | 5.7 | 0 | 0.114 | 0.09 | 0 | 0.52 |
| World Popula | TP-BERTa | 188 | 14 | 11 | 3 | 0.2 | 146.3 | 0 | 0.502 | 0.293 | 0 | 1 |
| train_0988_t | TP-BERTa | 158 | 11 | 10 | 1 | 0.1 | 3 | 0 | 0.335 | 0.16 | 0.016 | 0.881 |
| train_1660_S | TP-BERTa | 158 | 6 | 6 | 0 | 0 | 0 | 0 | 0.589 | 0.252 | 0 | 1 |
| train_1266_C | TP-BERTa | 156 | 12 | 12 | 0 | 0 | 0 | 0 | 0.104 | 0.168 | 0 | 1 |
| train_0125_p | TP-BERTa | 155 | 10 | 10 | 0 | 0 | 0 | 0 | 0.305 | 0.222 | 0 | 1 |
| train_1616_m | TP-BERTa | 119 | 4 | 3 | 1 | 0.2 | 3 | 0 | 0.505 | 0.289 | 0 | 1 |
| train_1755_D | TP-BERTa | 117 | 13 | 5 | 8 | 0.6 | 2.1 | 0 | 0.125 | 0.138 | 0 | 1 |
| train_0364_s | TP-BERTa | 116 | 6 | 6 | 0 | 0 | 0 | 0 | 0.365 | 0.217 | 0 | 1 |
| train_0225_v | TP-BERTa | 108 | 7 | 3 | 4 | 0.6 | 2.5 | 0 | 0.129 | 0.169 | 0 | 1 |
| train_0117_f | TP-BERTa | 99 | 4 | 4 | 0 | 0 | 0 | 0 | 0.262 | 0.182 | 0 | 0.976 |
| train_0261_a | TP-BERTa | 89 | 3 | 1 | 2 | 0.7 | 8 | 0 | 0.413 | 0.202 | 0 | 1 |

Table 12: TANGO graph head (regression): tuned hyperparameters.

| Hyperparameter | Distribution | Range / Values | Notes |
|---|---|---|---|
| topK (dynamic only) | integer | $[3, 20]$ | Capped at $\sqrt{N}$ |
| num_layers | integer | $[2, 4]$ | |
| hidden_channels | categorical | $\{64, 128, 256\}$ | |
| dropout | uniform | $[0.30, 0.70]$ | |
| lr_ft (joint only) | log-uniform | $[10^{-5}, 5 \times 10^{-4}]$ | |
| lr_gnn | log-uniform | $[10^{-4}, 10^{-2}]$ | |
| weight_decay | log-uniform | $[10^{-6}, 10^{-3}]$ | |
| loss | categorical | $\{\text{MSE, SmoothL1}\}$ | |
| Arch-specific | fixed/int | – | GAT/GATv2 heads $\in \{1, 2, 4\}$; TAGConv $K \in [3, 10]$; PNA {mean,max,min,std} |

Table 13: CatBoost (regression): search space and fixed knobs.

| Hyperparameter | Distribution | Range / Values |
|---|---|---|
| max_depth | integer | $[3, 10]$ |
| learning_rate | log-uniform | $[10^{-5}, 1]$ |
| bagging_temperature | uniform | $[0, 1]$ |
| l2_leaf_reg | log-uniform | $[1, 10]$ |
| leaf_estimation_iterations | integer | $[1, 10]$ |
| loss_function | fixed | RMSE |
| iterations | fixed | 2000 |
| random_seed | fixed | 42 |

# D APPENDIX: CLASSIFICATION RESULTS

This appendix reports the detailed dataset–by–dataset results for all classification benchmarks. For each dataset, we provide the Macro-F1 Scores. The best value(s) per dataset are highlighted in bold. All dataset names are truncated to the first 16 characters, and alphabetically ordered. Readers mainly interested in aggregated results and rankings may refer to the main paper.

Table 17: Classification results (Macro-F1). Best per row in **bold**.

| Dataset | CB | XGB | FT-T | TANGO | MITRA | TabICL | TabPFNv2 |
|---|---|---|---|---|---|---|---|
| Amazon_employee_ | 0.674 | 0.716 | 0.487 | 0.511 | 0.487 | **0.753** | 0.709 |
| anneal | **1.000** | **1.000** | 0.593 | 0.872 | 0.948 | **1.000** | **1.000** |
| APSFailure | **0.929** | 0.924 | 0.853 | 0.870 | 0.822 | 0.928 | |
| audit_data | **1.000** | **1.000** | **1.000** | **1.000** | **1.000** | **1.000** | **1.000** |
| audit_risk | **1.000** | **1.000** | **1.000** | **1.000** | **1.000** | **1.000** | **1.000** |
| b_depressed | 0.473 | 0.477 | 0.487 | **0.565** | 0.479 | 0.470 | 0.426 |
| bank-marketing | 0.603 | 0.609 | 0.668 | 0.668 | 0.657 | **0.673** | 0.655 |
| Bank_Customer_Ch | 0.743 | 0.743 | 0.755 | 0.769 | 0.747 | 0.773 | **0.777** |
| Bank_Personal_Lo | 0.545 | **0.566** | 0.532 | 0.560 | 0.534 | 0.532 | 0.542 |
| BankNoteAuthenti | **1.000** | **1.000** | **1.000** | **1.000** | **1.000** | **1.000** | **1.000** |
| blood-transfusio | 0.557 | 0.651 | 0.571 | 0.666 | 0.579 | **0.679** | 0.663 |
| Breast_Cancer | 0.759 | 0.766 | **0.776** | 0.775 | 0.752 | 0.766 | 0.776 |
| bt_dataset_t3 | 0.988 | **1.000** | **1.000** | **1.000** | **1.000** | | **1.000** |
| churn | 0.909 | 0.919 | 0.586 | **0.932** | 0.916 | 0.889 | 0.925 |
| coil2000_insuran | 0.482 | 0.548 | 0.483 | 0.584 | 0.583 | **0.616** | 0.551 |
| credit-g | 0.665 | **0.675** | 0.638 | 0.666 | 0.647 | 0.633 | 0.656 |
| credit_card_clie | 0.683 | 0.674 | 0.443 | 0.701 | 0.699 | **0.713** | 0.703 |
| Customer_Behavio | 0.916 | 0.916 | **0.973** | **0.973** | **0.973** | 0.918 | 0.945 |
| customer_satisfa | 0.958 | 0.957 | 0.961 | **0.962** | 0.926 | 0.959 | |
| diabetes | 0.732 | 0.739 | 0.368 | 0.679 | 0.795 | 0.701 | **0.824** |
| Diabetes130US | 0.485 | 0.541 | 0.562 | **0.579** | 0.498 | 0.556 | 0.559 |

*continued on next page*

| Dataset | CB | XGB | FT-T | TANGO | MITRA | TabICL | TabPFNv2 |
|---|---|---|---|---|---|---|---|
| diabetes_data_up | 0.979 | **1.000** | 0.916 | **1.000** | 0.936 | 0.979 | 0.979 |
| E-CommereShippin | **0.696** | 0.693 | 0.690 | 0.695 | 0.685 | 0.690 | 0.689 |
| Employee Satisfa | 0.408 | 0.458 | 0.435 | **0.572** | 0.494 | 0.391 | 0.324 |
| Fitness_Club | 0.691 | 0.673 | 0.600 | 0.710 | 0.654 | **0.719** | 0.697 |
| GiveMeSomeCredit | 0.630 | 0.628 | 0.683 | 0.686 | 0.689 | **0.694** | |
| hazelnut-spread- | 0.872 | 0.908 | 0.395 | 0.940 | 0.924 | **0.968** | 0.952 |
| heloc | 0.693 | 0.695 | 0.344 | 0.691 | 0.699 | 0.688 | **0.700** |
| HR_Analytics_Job | 0.738 | 0.731 | 0.748 | 0.758 | 0.752 | **0.762** | 0.762 |
| in_vehicle_coupo | 0.773 | 0.765 | 0.690 | 0.767 | 0.700 | **0.782** | 0.743 |
| Is-this-a-good-c | 0.465 | 0.540 | 0.468 | 0.612 | 0.618 | **0.644** | 0.621 |
| jm1 | 0.588 | 0.639 | 0.438 | 0.644 | 0.641 | **0.694** | 0.637 |
| kddcup09_appeten | 0.520 | 0.518 | 0.567 | 0.570 | 0.531 | **0.587** | 0.574 |
| loan_train | 0.678 | 0.684 | 0.647 | 0.707 | 0.689 | 0.699 | **0.733** |
| Marketing_Campai | 0.708 | 0.713 | 0.453 | 0.691 | 0.732 | 0.731 | **0.770** |
| maternal_health_ | **0.873** | 0.859 | 0.390 | 0.858 | 0.783 | 0.864 | 0.853 |
| MIC | 0.172 | 0.195 | 0.007 | 0.167 | 0.186 | **0.255** | 0.183 |
| NATICUSdroid | 0.957 | **0.958** | 0.941 | 0.946 | 0.936 | 0.949 | 0.954 |
| new_model | **1.000** | **1.000** | **1.000** | **1.000** | **1.000** | **1.000** | **1.000** |
| NFL | **1.000** | **1.000** | **1.000** | **1.000** | **1.000** | **1.000** | **1.000** |
| online_shoppers_ | 0.811 | 0.803 | 0.824 | **0.824** | 0.799 | 0.812 | 0.816 |
| piracydataset | 0.591 | 0.623 | **0.676** | **0.676** | 0.623 | 0.602 | 0.617 |
| polish_companies | 0.688 | 0.711 | 0.480 | 0.699 | 0.822 | 0.846 | **0.854** |
| qsar-biodeg | 0.884 | 0.825 | 0.689 | 0.858 | 0.878 | 0.872 | **0.885** |
| SDSS17 | 0.971 | 0.970 | 0.958 | 0.960 | 0.965 | **0.971** | |
| seismic-bumps | 0.476 | 0.565 | 0.477 | 0.477 | **0.677** | 0.645 | 0.628 |
| splice | 0.966 | **0.978** | 0.947 | 0.970 | 0.959 | 0.939 | 0.973 |
| students_dropout | 0.332 | 0.306 | 0.527 | 0.738 | 0.710 | 0.739 | **0.750** |
| taiwanese_bankru | 0.608 | 0.553 | 0.493 | 0.717 | 0.730 | 0.668 | **0.766** |
| train_0124_analc | 0.889 | 0.889 | **1.000** | **1.000** | 0.889 | 0.889 | 0.889 |
| train_0284_bank8 | 0.949 | 0.950 | 0.946 | **0.955** | 0.951 | 0.952 | 0.950 |
| train_0292_cpu_s | 0.922 | 0.919 | 0.909 | 0.923 | 0.910 | 0.921 | **0.924** |
| train_0312_cpu_a | 0.913 | 0.919 | 0.905 | 0.917 | 0.912 | 0.918 | **0.924** |
| train_0345_delta | 0.942 | 0.939 | 0.933 | 0.940 | 0.938 | 0.939 | **0.946** |
| train_0356_delta | **0.886** | 0.879 | 0.870 | 0.886 | 0.881 | 0.878 | 0.884 |
| train_0400_analc | **1.000** | 0.997 | 0.991 | 0.997 | 0.997 | 0.994 | 0.997 |
| train_0406_visua | 0.679 | 0.585 | 0.679 | **1.000** | 0.679 | 0.679 | 0.679 |
| train_0408_phary | 0.734 | 0.679 | 0.585 | **0.778** | 0.679 | 0.679 | 0.734 |
| train_0419_pm10 | 0.653 | 0.483 | 0.601 | **0.660** | 0.559 | 0.582 | 0.531 |
| train_0424_autoP | 0.844 | 0.844 | 0.844 | **0.928** | 0.844 | 0.844 | 0.844 |
| train_0435_strik | 0.984 | **1.000** | 0.938 | 0.984 | **1.000** | 0.984 | **1.000** |
| train_0437_quake | 0.501 | 0.489 | 0.516 | **0.583** | 0.554 | 0.572 | 0.503 |
| train_0445_arsen | **0.829** | **0.829** | **0.829** | **0.829** | **0.829** | **0.829** | **0.829** |
| train_0446_arsen | 0.580 | 0.677 | 0.630 | **0.760** | 0.608 | 0.719 | 0.625 |
| train_0446_newto | **0.923** | **0.923** | **0.923** | **0.923** | 0.845 | 0.845 | 0.845 |
| train_0447_arsen | **0.492** | **0.492** | **0.492** | **0.492** | **0.492** | **0.492** | **0.492** |
| train_0472_analc | 0.418 | 0.451 | 0.513 | **0.677** | 0.439 | 0.439 | 0.439 |
| train_0509_polle | **0.538** | 0.500 | 0.530 | 0.514 | 0.496 | 0.474 | 0.495 |
| train_0526_colle | 0.970 | 0.951 | 0.971 | **0.990** | 0.980 | 0.961 | 0.971 |
| train_0541_plasm | 0.597 | 0.542 | 0.441 | **0.625** | 0.583 | 0.600 | 0.400 |
| train_0546_analc | **1.000** | **1.000** | **1.000** | **1.000** | **1.000** | **1.000** | **1.000** |
| train_0555_socmo | **0.964** | 0.947 | 0.883 | 0.950 | 0.947 | 0.935 | 0.916 |
| train_0885_compa | 0.667 | 0.679 | 0.673 | **0.688** | 0.679 | 0.675 | 0.680 |
| train_0948_Ishwa | 0.976 | 0.972 | 0.921 | 0.948 | 0.976 | 0.981 | **0.986** |
| train_1006_Titan | 0.685 | 0.685 | 0.677 | **0.689** | 0.676 | 0.676 | 0.676 |
| train_1011_cleve | 0.791 | 0.861 | 0.724 | **0.862** | 0.861 | 0.824 | 0.861 |
| train_1142_Sick_ | 0.978 | 0.977 | 0.924 | **0.989** | 0.904 | 0.936 | 0.966 |
| train_1201_Gende | 0.988 | 0.985 | 0.982 | 0.988 | 0.988 | **0.991** | 0.988 |
| train_1333_ricci | 0.890 | **1.000** | **1.000** | **1.000** | 0.890 | 0.890 | 0.890 |
| train_1408_natio | **1.000** | **1.000** | **1.000** | **1.000** | **1.000** | **1.000** | **1.000** |
| train_1413_shill | **1.000** | **1.000** | 0.982 | 0.996 | **1.000** | **1.000** | **1.000** |
| train_1451_early | **1.000** | **1.000** | **1.000** | **1.000** | 0.943 | 0.962 | 0.981 |

| Dataset | CB | XGB | FT-T | TANGO | MITRA | TabICL | TabPFNv2 |
|---|---|---|---|---|---|---|---|
| train_1458_kdd_i | 0.870 | 0.866 | 0.832 | 0.865 | **0.876** | 0.868 | 0.860 |
| train_1461_heart | 0.741 | 0.718 | 0.741 | **0.758** | 0.718 | 0.718 | 0.718 |
| train_1512_eye_m | 0.639 | **0.687** | 0.633 | 0.658 | 0.576 | 0.639 | 0.647 |
| train_1564_Mammo | 0.804 | 0.792 | 0.852 | **0.865** | 0.829 | 0.865 | 0.853 |
| train_1578_kdd_i | 0.897 | **0.899** | 0.868 | 0.891 | 0.889 | 0.893 | 0.895 |
| train_1592_Diabe | 0.686 | 0.738 | 0.750 | **0.824** | 0.601 | 0.704 | 0.692 |
| train_1600_VulNo | **0.499** | 0.497 | **0.499** | **0.499** | **0.499** | **0.499** | 0.499 |
| train_1619_NBA-2 | **1.000** | **1.000** | **1.000** | **1.000** | **1.000** | **1.000** | **1.000** |
| train_1635_Is-th | 0.559 | 0.546 | 0.675 | 0.713 | 0.636 | 0.657 | **0.715** |
| train_1692_Gende | 0.970 | 0.960 | 0.966 | **0.972** | 0.952 | 0.954 | 0.948 |
| train_1736_combi | 0.994 | 0.996 | 0.998 | **1.000** | 0.996 | **1.000** | 0.996 |
| train_1742_Loan- | 0.661 | 0.661 | 0.661 | **0.676** | 0.661 | 0.646 | 0.651 |
| train_1752_Wisco | 0.984 | 0.984 | 0.984 | **1.000** | 0.984 | 0.967 | 0.984 |
| train_1759_Red– | 0.994 | 0.998 | **1.000** | **1.000** | 0.996 | **1.000** | 0.996 |
| train_1774_Early | 0.961 | **1.000** | **1.000** | **1.000** | 0.962 | 0.962 | 0.981 |
| train_1898_Perso | 0.565 | 0.584 | 0.576 | **0.595** | 0.567 | 0.588 | 0.564 |
| train_2304_elect | 0.736 | 0.722 | 0.756 | 0.773 | 0.751 | 0.755 | **0.822** |
| train_2305_elect | 0.764 | 0.794 | 0.773 | 0.804 | 0.841 | 0.817 | **0.851** |
| train_2306_elect | 0.751 | 0.736 | 0.699 | 0.775 | 0.766 | 0.751 | **0.799** |
| train_2308_elect | 0.799 | 0.794 | 0.701 | 0.779 | 0.798 | 0.773 | **0.847** |
| train_2389_airli | 0.536 | 0.534 | 0.551 | 0.593 | 0.617 | 0.570 | **0.617** |
| train_2390_airli | 0.562 | 0.564 | 0.548 | **0.594** | 0.579 | 0.540 | 0.508 |
| train_2391_airli | 0.586 | 0.583 | 0.501 | 0.607 | 0.583 | 0.558 | **0.608** |
| train_2392_airli | 0.540 | 0.530 | 0.581 | **0.615** | 0.606 | 0.591 | 0.589 |
| train_2393_airli | 0.571 | 0.530 | 0.544 | **0.631** | 0.627 | 0.617 | 0.607 |
| train_2619_sf-po | 0.472 | 0.485 | 0.548 | **0.594** | 0.472 | 0.472 | 0.485 |
| train_2620_sf-po | 0.470 | 0.505 | 0.523 | **0.570** | 0.451 | 0.472 | 0.465 |
| train_2621_sf-po | 0.500 | 0.540 | 0.467 | **0.564** | 0.472 | 0.536 | 0.472 |
| train_2622_sf-po | 0.472 | 0.467 | 0.502 | **0.540** | 0.472 | 0.472 | 0.472 |
| train_2703_compa | 0.698 | **0.707** | 0.683 | 0.697 | 0.704 | 0.698 | 0.700 |
| TravelInsuranceP | **0.811** | 0.799 | 0.776 | 0.789 | 0.805 | 0.808 | 0.799 |
| trial | **1.000** | **1.000** | **1.000** | **1.000** | **1.000** | **1.000** | **1.000** |
| UniversalBank | 0.535 | 0.545 | 0.537 | **0.570** | 0.555 | 0.532 | 0.542 |
| website_phishing | 0.861 | 0.866 | 0.852 | 0.872 | 0.871 | **0.903** | 0.888 |

Table 14: XGBoost (regression): search space and fixed knobs.

| Hyperparameter | Distribution | Range / Values |
|---|---|---|
| max_depth | integer | $[3, 10]$ |
| min_child_weight | log-uniform | $[10^{-8}, 10^5]$ |
| subsample | uniform | $[0.5, 1.0]$ |
| learning_rate | log-uniform | $[10^{-5}, 1.0]$ |
| colsample_bylevel/bytree | uniform | $[0.5, 1.0]$ |
| gamma | log-uniform | $[10^{-8}, 10^2]$ |
| $\lambda$ (L2) / $\alpha$ (L1) | log-uniform | $[10^{-8}, 10^2]$ |
| n_estimators | fixed | 2000 |
| objective | fixed | reg:squarederror |
| random_state | fixed | 42 |

# E   APPENDIX: REGRESSION RESULTS

This appendix reports detailed dataset–by–dataset results for all regression benchmarks. For each dataset, we provide the RMSE on the test split (lower is better throughout). The best value(s) per dataset are highlighted in bold. Datasets' names are truncated to the first 16 characters, and are listed in alphabetical order. The table is intended for completeness and reproducibility; aggregate summaries (rank-1 wins, average rank, and average relative gap) are given in the main paper.

Table 18: Classification results (RMSE). Best per row in **bold**.

| Dataset | CB | XGB | FT-T | TANGO | MITRA | TabPFNv2 |
|---|---|---|---|---|---|---|
| airfoil_self_noi | 0.232 | 0.200 | 0.234 | 0.159 | 0.197 | **0.156** |
| Another-Dataset- | 0.357 | 0.339 | 0.358 | 0.349 | 0.340 | **0.328** |
| concrete_compres | 0.255 | 0.228 | 0.260 | 0.215 | 0.230 | **0.186** |
| diamonds | 0.129 | 0.121 | 0.140 | 0.122 | 0.151 | **0.117** |
| Food_Delivery_Ti | 0.786 | **0.784** | 0.856 | 0.838 | 0.839 | 0.810 |
| healthcare_insur | 0.472 | 0.462 | 0.502 | 0.462 | **0.457** | 0.467 |
| houses | 0.377 | 0.362 | 0.380 | 0.373 | 0.397 | **0.340** |
| mechanical_analy | **0.000** | 0.002 | 0.257 | 0.009 | 0.041 | 0.002 |
| miami_housing | 0.266 | 0.259 | 0.284 | 0.261 | 0.269 | **0.246** |
| ph-data | 0.075 | 0.081 | 0.069 | 0.069 | **0.066** | 0.070 |
| physiochemical_p | 0.591 | 0.547 | 0.580 | 0.546 | 0.688 | **0.530** |
| QSAR_fish_toxici | 0.607 | 0.623 | 0.670 | 0.621 | 0.611 | **0.606** |
| real_estate_list | 0.005 | 0.003 | 0.004 | 0.002 | 0.003 | **0.001** |
| superconductivit | 0.377 | **0.348** | 0.403 | 0.389 | 0.460 | 0.357 |
| thyroidDF | **0.215** | 0.219 | 0.236 | 0.232 | 0.234 | 0.228 |
| train_0117_fruit | 0.320 | 0.259 | 0.293 | **0.243** | 0.264 | 0.267 |
| train_0125_phary | 0.190 | 0.157 | **0.129** | 0.131 | 0.131 | 0.130 |
| train_0130_breas | 0.207 | 0.205 | 0.215 | **0.199** | 0.206 | 0.209 |
| train_0149_socmo | 0.034 | 0.035 | 0.038 | **0.025** | 0.029 | 0.034 |
| train_0211_analc | 0.109 | 0.086 | 0.086 | **0.081** | 0.085 | 0.082 |
| train_0225_veter | 0.143 | **0.094** | 0.144 | 0.101 | 0.101 | 0.099 |
| train_0226_analc | 0.064 | 0.078 | 0.078 | 0.064 | **0.061** | 0.065 |
| train_0235_plasm | 0.141 | 0.132 | 0.140 | **0.131** | 0.136 | 0.132 |
| train_0237_arsen | **0.002** | 0.045 | 0.011 | 0.007 | 0.029 | 0.040 |
| train_0251_arsen | **0.003** | 0.045 | 0.049 | 0.004 | 0.052 | 0.038 |
| train_0258_no2 | 0.106 | 0.105 | 0.111 | **0.103** | 0.104 | 0.105 |
| train_0259_strik | 0.142 | 0.140 | 0.145 | 0.140 | 0.139 | **0.139** |
| train_0261_analc | 0.203 | 0.171 | 0.182 | **0.107** | 0.192 | 0.195 |
| train_0272_kdd_c | 0.159 | 0.152 | 0.146 | **0.109** | 0.124 | 0.144 |
| train_0273_kdd_c | 0.140 | **0.129** | 0.156 | 0.132 | 0.129 | 0.130 |
| train_0274_kdd_c | 0.102 | 0.098 | 0.126 | **0.097** | 0.109 | 0.110 |
| train_0279_kdd_c | 0.134 | 0.127 | 0.147 | 0.131 | **0.120** | 0.136 |
| train_0364_sleut | 0.125 | 0.120 | 0.185 | **0.099** | 0.132 | 0.119 |

*continued on next page*

| Dataset | CB | XGB | FT-T | TANGO | MITRA | TabPFNv2 |
|---|---|---|---|---|---|---|
| train_0907_UCI-s | 0.076 | 0.082 | 0.075 | **0.059** | 0.074 | 0.085 |
| train_0911_fores | 0.172 | 0.170 | 0.171 | **0.168** | 0.172 | 0.172 |
| train_0925_Concr | 0.040 | 0.042 | 0.051 | 0.042 | 0.043 | **0.039** |
| train_0988_test_ | 0.175 | 0.159 | 0.164 | **0.133** | 0.149 | 0.141 |
| train_1065_hunga | 0.051 | 0.055 | 0.076 | 0.052 | 0.051 | **0.050** |
| train_1118_jura | 0.058 | 0.061 | 0.045 | **0.041** | 0.047 | 0.057 |
| train_1222_premi | 0.740 | 0.744 | 0.802 | 0.737 | 0.728 | **0.724** |
| train_1228_Premi | 0.101 | 0.096 | 0.104 | 0.104 | 0.090 | **0.077** |
| train_1260_optic | **0.000** | 0.000 | 0.000 | 0.000 | 0.001 | 0.002 |
| train_1266_CSM | 0.107 | 0.123 | 0.112 | 0.114 | **0.093** | 0.105 |
| train_1267_autoM | 0.083 | 0.087 | 0.080 | **0.078** | 0.081 | 0.078 |
| train_1331_datas | 0.177 | **0.177** | 1.180 | 0.177 | 0.178 | 0.181 |
| train_1417_ibm-e | 0.098 | 0.098 | 0.114 | 0.099 | 0.095 | **0.092** |
| train_1449_garme | 0.163 | **0.134** | 0.194 | 0.134 | 0.140 | 0.140 |
| train_1464_dow-j | 0.143 | 0.147 | 0.024 | 0.024 | 0.021 | **0.007** |
| train_1528_18Pro | 0.152 | 0.134 | 0.176 | **0.131** | 0.140 | 0.140 |
| train_1564_Concr | 0.053 | 0.053 | 0.069 | 0.055 | 0.060 | **0.046** |
| train_1591_Super | 0.027 | 0.026 | 0.019 | **0.017** | 0.029 | 0.026 |
| train_1594_Spoti | 0.134 | 0.134 | 0.151 | 0.128 | 0.137 | **0.124** |
| train_1616_myiri | 0.148 | 0.117 | 0.121 | **0.103** | 0.103 | 0.104 |
| train_1623_GameS | 0.006 | 0.004 | 0.005 | 0.004 | 0.006 | **0.003** |
| train_1624_Alcoh | 0.053 | 0.055 | 0.041 | **0.038** | 0.053 | 0.041 |
| train_1640_Calcu | 0.041 | 0.040 | 0.071 | 0.045 | 0.046 | **0.039** |
| train_1660_Swiss | 0.095 | 0.098 | 0.098 | **0.060** | 0.073 | 0.070 |
| train_1712_Runni | 0.016 | 0.014 | 0.017 | 0.014 | 0.013 | **0.009** |
| train_1755_Detai | 0.071 | 0.049 | 0.090 | **0.040** | 0.059 | 0.051 |
| train_1769_Faceb | 0.036 | 0.035 | 0.036 | 0.034 | 0.032 | **0.029** |
| train_1787_Lisbo | 0.039 | 0.047 | 0.073 | 0.035 | **0.029** | 0.032 |
| train_1845_Predi | 0.134 | 0.130 | 0.144 | **0.121** | 0.131 | 0.131 |
| train_1848_Minne | 0.001 | **0.000** | 0.014 | 0.001 | 0.025 | 0.024 |
| train_1869_Bitco | 0.027 | 0.024 | 0.022 | **0.020** | 0.023 | 0.021 |
| train_1872_Fores | **0.000** | 0.001 | 0.008 | 0.001 | 0.024 | 0.041 |
| train_1874_Goodr | 0.070 | 0.055 | **0.045** | 0.045 | 0.053 | 0.082 |
| train_1878_COVID | 0.012 | 0.018 | **0.010** | 0.013 | 0.016 | 0.091 |
| train_1890_ECDC- | 0.003 | 0.003 | 0.006 | **0.001** | 0.030 | 0.071 |
| train_1900_Anoth | 0.083 | 0.078 | 0.104 | 0.083 | 0.079 | **0.074** |
| train_1901_Netfl | **0.053** | 0.055 | 0.061 | 0.061 | 0.061 | 0.054 |
| train_2168_Inter | 0.181 | 0.184 | 0.194 | **0.176** | 0.183 | 0.183 |
| train_2644_concr | **0.038** | 0.041 | 0.054 | 0.047 | 0.044 | 0.040 |
| User Knowledge | 0.239 | 0.237 | 0.232 | 0.233 | **0.195** | 0.198 |
| wine_quality | 0.747 | **0.679** | 0.802 | 0.781 | 0.764 | 0.821 |
| wines_SPA | 0.018 | 0.075 | **0.003** | 0.006 | 0.057 | 0.140 |
| World Population | 0.017 | 0.005 | 0.053 | 0.046 | 0.006 | **0.004** |

## USE OF LARGE LANGUAGE MODELS (LLMS)

Large Language Models (LLMs) were used solely to assist with table formatting. All ideas, scientific content, analyses, and results are the authors' own.

