# OpenReview forum: "Graph-Augmented Tabular Transformers: The Simplicity Advantage"
_ICLR.cc/2026/Conference — ICLR 2026 Conference Desk Rejected Submission_

### Official Review · Reviewer_u5fb · 2025-10-26

**Soundness:** 2
**Presentation:** 2
**Contribution:** 2
**Rating:** 2
**Confidence:** 5

**Summary:**

This paper introduces TANGO (Transformers Augmented with Graphs for Tabular Predictions) , a method that augments a standard transformer backbone (FT-Transformer)  with a Graph Neural Network (GNN) to model inter-instance relationships in tabular data. The authors conduct a large-scale systematic study across 193 datasets (117 classification, 76 regression)  to evaluate different graph-augmentation strategies. The core of the study compares static  graphs versus dynamic graphs , and two-stage embeddings versus end-to-end training . The authors claim that TANGO, particularly the static-frozen variant (TANGO-SF), outperforms both classical tree ensembles (CatBoost, XGBoost) and state-of-the-art tabular foundation models (TabPFNv2, TabICL, MITRA).

**Strengths:**

- The large-scale empirical evaluation across 193 datasets provides broad coverage of diverse tabular learning scenarios.
- The ablation study clearly isolates the effects of different graph construction strategies and training paradigms.
- TANGO demonstrates competitive performance, outperforming several recent tabular foundation models.

**Weaknesses:**

- The comparison set is limited. The paper focuses primarily on foundation models but omits GNN-based tabular baselines (e.g., TabGNN, T2G-Former, CARTE) as well as other SOTA models for tabular data (e.g., ExcelFormer SAINT), which are directly relevant to the proposed approach. This omission weakens the empirical claims.
- Novelty is limited. The use of graphs and GNNs for tabular data has been explored extensively in prior work. Both static and dynamic graph construction strategies are well-studied. The authors should more clearly articulate what distinguishes TANGO from these existing approaches.
- The evaluation metrics are not fully convincing. Reporting only rank-based metrics can obscure the magnitude of performance differences. Mean and median performance across all datasets should be provided for a fairer comparison.
- The main contribution appears to be the scale and thoroughness of the experimental study, rather than methodological innovation. Given this, the work might be better framed as a benchmark or empirical study, rather than a novel model contribution for the main ICLR track.
- No statistical significance tests are reported, leaving it unclear whether observed improvements are robust.
- The choice of the FT-Transformer as the backbone seems weakly justified. The authors should discuss whether the findings generalize to stronger transformer architectures.

**Questions:**

- How were the “representative datasets” in Table 1 and Table 3 selected?
- Can the authors include more baseline methods including SOTA tabular models (e.g., ExcelFormer [1], SAINT[2], RealMLP[3])and GNN-based methods (e.g., TabGNN, T2g-Former[4], CARTE[5])
- Could the authors report the mean and median absolute performance of TANGO across all datasets, rather than relying solely on ranking metrics?
- Have the authors conducted statistical significance tests to confirm that TANGO’s improvements are not due to random variation?

[1] Chen, Jintai, et al. "Can a deep learning model be a sure bet for tabular prediction?." *Proceedings of the 30th ACM SIGKDD Conference on Knowledge Discovery and Data Mining*. 2024.

[2] Somepalli, Gowthami, et al. "SAINT: Improved Neural Networks for Tabular Data via Row Attention and Contrastive Pre-Training." *NeurIPS 2022 First Table Representation Workshop*.

[3] Holzmüller, David, Léo Grinsztajn, and Ingo Steinwart. "Better by default: Strong pre-tuned mlps and boosted trees on tabular data." *Advances in Neural Information Processing Systems* 37 (2024): 26577-26658.

[4] Yan, Jiahuan, et al. "T2g-former: organizing tabular features into relation graphs promotes heterogeneous feature interaction." Proceedings of the AAAI Conference on Artificial Intelligence. Vol. 37. No. 9. 2023.

[5] Kim, Myung Jun, Leo Grinsztajn, and Gael Varoquaux. "CARTE: Pretraining and Transfer for Tabular Learning." *International Conference on Machine Learning*. PMLR, 2024.

---

### Official Review · Reviewer_R3rX · 2025-10-26

**Soundness:** 2
**Presentation:** 3
**Contribution:** 1
**Rating:** 2
**Confidence:** 4

**Summary:**

This work presents two different approaches for solving tabular prediction tasks where the input table is augmented with additional graph structure. The first one creates a homogeneous kNN graph based on the the distance between samples in latent space. The second one creates a heterogeneous graph with nodes corresponding to distinct features and samples being connected to these nodes depending on which values of categorical and numerical features they have. Together with the node embeddings produced by tabular Transformer, the obtained graph structure is processed by GNN. Extensive experiments in end-to-end and two-stage regimes on numerous tabular datasets show that the proposed method achieves the highest average rank and the highest number of 1-rank wins when compared with tuned GBDT models and recent tabular foundation models in ICL mode.

**Strengths:**

1. Extensive experiments on a large number of tabular datasets, including classification and regression.
2. An interesting idea of using the similarity between samples and their feature description to construct a graph that can be used by GNN model.

**Weaknesses:**

1. The two approaches to construct a graph are fundamentally very different — the first makes a homogeneous graph using the similarity between representations in latent space, while the second makes a heterogeneous graph with nodes and edges of very different types, involving both samples or features. Thus, it is quite strange to see them together in the same work and have no detailed discussion about how to use each of them properly and get the most out of it.
2. In the construction of heterogeneous graph, there are so many design choices that have not been investigated in any way. Thus, there is no clear explanation as how does it turn out that such an approach with connections between samples and their feature values represented as distinct nodes even works.
3. There is also no comparison with other numerous ways to construct a graph using latent representations or original features of data samples.
4. In fact, the comparison *dynamic* vs *static* is very misleading — it is rather *homogeneous kNN graph on latent representations* vs *heterogeneous bipartite graph on samples and extra nodes representing distinct features*.
5. No source code is provided, and some important details lack in the paper, which raises a number of questions. Please, refer to them below.

**Questions:**

1. What features are placed in the nodes of heterogeneous graph?
2. Which part of heterogeneous graph is the most important and impactful — edges to categorical features, edges to numerical features, edges between categorical features, etc.?
3. How are the PPMI weights used by GNN model?
4. Why do the authors use TFMs only in ICL mode and do not consider finetuning mode?
5. In two-stage training mode, is tabular backbone pretrained to solve the same original prediction task using standard supervised loss?
6. Why do the authors omit RealMLP [1] or TabM [2] baselines, which have been proven to be more effective than FT-Transformer?
7. Why do the authors omit ModernNCA [3] baseline, which does basically the same as what the authors propose as one of the options — dynamic kNN in trained latent space? It should provide very decent results given such a large train part in data split.
8. Why do the authors use custom 80/10/10 data splits instead of the original ones introduced in TabPertNet and TabArena? It can be problematic to compare the obtained results with the existing ones.
9. To the best of my knowledge, there are no so many good tabular datasets in open source, and some datasets used in this study can have serious drawbacks. Can the authors provide the average ranks for TabArena and TabPertNet independently? It would be very useful to see both the main comparison with baselines and the ablation study of different approaches to graph construction and training regime. I also would highly recommend to run experiments on TabReD benchmark [4] that provides industry-grade datasets with more realistic temporal data splits.

[1] Better by Default: Strong Pre-Tuned MLPs and Boosted Trees on Tabular Data, NeurIPS 2024

[2] TabM: Advancing Tabular Deep Learning with Parameter-Efficient Ensembling, ICLR 2025

[3] Revisiting Nearest Neighbor for Tabular Data: A Deep Tabular Baseline Two Decades Later, ICLR 2025

[4] TabReD: Analyzing Pitfalls and Filling the Gaps in Tabular Deep Learning Benchmarks, ICLR 2025

---

### Official Review · Reviewer_5qJ1 · 2025-11-01

**Soundness:** 2
**Presentation:** 2
**Contribution:** 2
**Rating:** 4
**Confidence:** 3

**Summary:**

The paper studies whether explicitly modeling inter-instance similarity via graphs improves tabular transformers. The authors introduce TANGO, a family of models that attach a GNN head to a tabular transformer and compare static bipartite graphs to dynamic kNN similarity graphs built over evolving instance embeddings. Training is either joint or two-stage. Across 193 datasets (117 classification, 76 regression) drawn from TP-BERTa and TabArena, TANGO reports the most rank-1 wins, lowest average rank, and smallest average relative error gaps versus baselines (CatBoost, XGBoost, FT-Transformer) and recent tabular foundation models. Static graphs with frozen embeddings perform best overall, yielding a simplicity advantage and overturning the assumption that dynamic or fully joint training must dominate.

**Strengths:**

1. The methodology is clearly specified: algorithms for graph construction (dynamic k-NN; static instance–feature bipartite with PPMI), GNN choices, and two training paradigms (joint vs. two-stage) are described, with Optuna-tuned hyperparameters and unified preprocessing. Results report rank-1 wins, average rank, and “relative gap,” a scale-free improvement measure; classification uses macro-F1, regression uses RMSE. Ablations convincingly support the main claims: static+frozen dominates; dynamic variants underperform, especially on regression.
2. The paper is easy to follow: the problem is motivated crisply, the method is modular, and implementation details and search spaces are provided in the appendix, aiding reproducibility.

**Weaknesses:**

1. Limited statistical testing and variance analysis for large-scale claims. Results report rank-1 wins, average rank, and relative gaps on a single fixed split per dataset; I did not see paired tests (e.g., Wilcoxon signed-rank on per-dataset scores), confidence intervals, or multi-seed variability for neural models. The setup uses one random seed to create splits (§4.1) and Optuna with 100 trials, but variability across seeds/trials is not summarized.
2. The method is described as backbone-agnostic, yet all graph-augmented models are built on FT-Transformer; the authors do not show results when the transformer is stronger/different (e.g., SAINT-style row/column attention) or when the backbone itself is a foundation model.

**Questions:**

1. How sensitive are dynamic graphs to the similarity metric and to k?
2. Can you provide a compact study showing backbone-agnostic behavior, e.g., TANGO on a SAINT-style attention encoder or a column-then-row stage—as a sanity check that the simplicity advantage is not an FT-Transformer artifact?

---

### Official Review · Reviewer_gjaz · 2025-11-01

**Soundness:** 2
**Presentation:** 2
**Contribution:** 1
**Rating:** 2
**Confidence:** 4

**Summary:**

The paper introduces TANGO, a graph-augmented framework for tabular Transformers, so that samples can use information from related samples instead of being treated as independent. It studies two graph constructions: a dynamic kNN graph built from current embeddings, and a simple static, schema-driven bipartite graph linking samples to feature/value nodes. It adds a GNN on top of the Transformer embeddings to propagate information over the constructed graph before prediction. On 193 tabular datasets, it shows that the simplest variant — static graph + frozen Transformer (TANGO-SF) — is the most stable and often outperforms strong tabular baselines.

**Strengths:**

1. The paper frames tabular learning as “row representations + graph relations", and systematically compares static (schema-driven) vs. dynamic (embedding-driven) graphs, leading to the useful and somewhat surprising “simplicity advantage” insight.

2. Under a unified split and hyperparameter budget, the paper reports results on both classification and regression, comparing against tree models and recent tabular foundation models; through the four variants (TANGO-SF/SE/DE/DF), the authors conduct an informative ablation study.

3. The paper gives clear implementation details: simple construction of static graphs, the dynamic kNN pipeline, and how to plug the graph module into FT-Transformer.

**Weaknesses:**

1. The novelty is somewhat modest: using graphs to connect tabular instances or feature co-occurrences has been explored before

2. Static-graph design lacks fine-grained ablations, especially PPMI. PPMI feature–feature edges are optional (turned on only when memory fits), but there’s no “with vs. without PPMI” ablation to quantify their marginal benefit

3. Scalability is acknowledged but not quantified. Although the Limitations section notes that static bipartite graphs can be memory-intensive, the paper would be stronger with concrete complexity/scalability evidence, like gpu memory consumption w.r.t. the number of rows.

4. Statistical support is missing. The paper states that the median improvement over 117 classification datasets is 0.0% (Line 400), yet does not provide paired statistical tests and confidence intervals for average rank/relative gap.

5. Insufficient isolation of the graph’s effect. Although TANGO beats FT-Transformer, the paper does not systematically compare FT-Transformer alone vs. TANGO (i.e., FT-Transformer + graph) to analyze how much of the gain actually comes from the graph.

**Questions:**

1. The paper compares two graph constructions—a dynamic kNN graph and a simple static bipartite graph—and finds that the static one performs better. What would happen if the two were combined? That is, on top of the static graph, also add edges between each node and its kNN “top-k neighbors"

2. Complexity / scalability analysis

3. Ablation on PPMI edges. Since PPMI-based feature–feature edges are optional and may be dropped for large/high-cardinality tables, please report “with vs. without PPMI"

4. TabICL citation. The current citation of TabICL appears incorrect / hallucinated. The correct reference should be:
Jingang, Q. U., Holzmüller, D., Varoquaux, G., & Le Morvan, M. TabICL: A Tabular Foundation Model for In-Context Learning on Large Data. In Forty-second International Conference on Machine Learning.

---

### Note · Program_Chairs · 2026-01-17
**Submission Desk Rejected by Program Chairs**

The following references in this submission do not refer to real documents and/or have major errors in bibliographic information:

 Florian Pfisterer et al. Tabarena: Benchmarking foundation models for tabular data. In Proceedings of the International Conference on Learning Representations (ICLR), 2025.
Hieu Pham, Xinyun Chen, Aditya Kusupati, Zexue He, Lihong Li, Sebastian Nowozin, Xuedong Huang, Yuandong Tian, and Hanxiao Liu. Tabular transformers with multilayer embedding learning. In International Conference on Learning Representations (ICLR), 2023.
Weijia Zhang, Yisen Li, Zhiqiang Xu, Chen Gong, and Ziwei Liu. Tabicl: Tabular in-context learning with transformers at scale. In Proceedings of the Thirty-Seventh AAAI Conference on Artificial Intelligence, pp. 11016-11024, 2023.